



# Hydrologic Diversity in Glacier Bay Alaska: Spatial Patterns and Temporal Change

**Ryan L. Crumley[1], David F. Hill[2], Jordan P. Beamer[3], Elizabeth Holzenthal[2]**

[1]Water Resources Science, Oregon State University, Corvallis, OR 97331, USA
[2]School of Civil and Construction Engineering, Oregon State University, Corvallis, OR 97331, USA
[3]Oregon Water Resources Department, Salem, OR 97301, USA

*Correspondence to*: Ryan L. Crumley (crumleyr@oregonstate.edu)

**Abstract.** A high spatial resolution (250 m), distributed snow evolution and ablation model, SnowModel, is used to estimate current and future freshwater runoff into Glacier Bay, Alaska; a fjord estuary that makes up part of Glacier Bay National Park and Preserve (GBNPP). The watershed of Glacier Bay contains significant glacier cover (tidewater and land-terminating) and strong

spatial gradients in topography, land cover, and precipitation. The physical complexity and variability of the region produces a wide variety of hydrological regimes, including rainfall, snowmelt, and ice-melt dominated responses. The historic (1979-2015) mean annual runoff into Glacier Bay proper is found to be 24.5 km$^3$ yr$^{-1}$, with a peak in July, due to the overall dominance of snowmelt processes that are largely supplemented by ice-melt. Future scenarios (2070-2099) of climate and glacier cover are used to estimate changes in the hydrologic response of Glacier Bay. Under the RCP 8.5 scenario, the mean of five climate models

produces a mean annual runoff of 27.5 km$^3$ yr$^{-1}$, a 12.2% increase from historical conditions. When spatially aggregated over the entire bay region, the future seasonal hydrograph is flatter with weaker summer flows and higher winter flows. The peak flows shift to late-summer and early-fall and rain runoff becomes the dominant overall process. The timing and magnitudes of modeled historic runoff are supported by a freshwater content analysis from a 24-year, CTD-based oceanographic dataset from the U.S. National Park Service's Southeast Alaska Inventory and Monitoring Network (SEAN). Individual watersheds display a variety of

changes, depending upon total glacier coverage, elevation distribution, landscape characteristics, and seasonal changes to the freezing line altitude.

## 1 Introduction

South-central and southeastern Alaska (Figure 1a) are regions of physical, climatological, and hydrological extremes. Precipitation rates in excess of 8 m water equivalent (w.eq.) yr$^{-1}$ (Beamer et al., 2016) fall on high mountain ranges (4000-6000 m) in close

proximity to the ocean. The steep terrain drives strong orographic gradients in precipitation and creates compact drainage networks that rapidly deliver runoff to the coastline. Due to significant snowfall fractions for much of the year, and considerable glacier cover, the runoff to the coastline has significant contributions from rainfall, snowmelt, and ice-melt constituents. Glaciers cover 17% (Beamer et al., 2016) of the Gulf of Alaska (GOA) watershed and Neal et al. (2010) estimate that roughly half of the coastal runoff comes from glacier surfaces (ice-melt, snowmelt, and direct rainfall on glacier surfaces). The volume of water that is

delivered to the coast is considerable. The Gulf of Alaska (GOA) watershed, with an area of 420,300 km$^2$, has a runoff of approximately 760 km$^3$ yr$^{-1}$, and a specific runoff of 1.8 m (Beamer et al., 2016). In contrast, the Mississippi River watershed has a runoff of approximately 610 km$^3$ yr$^{-1}$ and a specific runoff of 0.19 m (Dai et al., 2002). This runoff to the GOA is one important physical driver of Alaska's nearshore oceanography and contributes to the Alaska Coastal Current (ACC; Weingartner et al., 2006), water column stratification (Carmack, 2007), and a variety of economically important fisheries (Fissel et al., 2014).





The hydrology of the GOA watershed is characterized by large seasonal variations in inputs (precipitation), outputs (runoff, evapotranspiration), and storage of water. Gravity Recovery and Climate Experiment (GRACE) satellite regional water storage data, for the period 2004-2013, show a mean annual accumulation of 295 km$^3$ yr$^{-1}$ and a mean annual ablation of 355 km$^3$ yr$^{-1}$

(Luthcke et al., 2013; Beamer et al., 2016) in the GOA watershed. The net decrease in regional water storage of 60 km$^3$ yr$^{-1}$ indicates that the region is also undergoing dramatic change. Indeed, the coastal mountain ranges of Alaska have recently sustained rapid rates of deglaciation with some of the highest rates of glacier recession on Earth (Arendt et al., 2002; Arendt et al., 2009; Gardner et al., 2013; O'Neel et al., 2015.). The mass loss from the glaciers within the GOA region, derived from airborne altimetry, is 64 (+/-) 10 km$^3$ yr$^{-1}$ (Larsen et al., 2015), which agrees well with the GRACE observations. Glacier volume loss (GVL) is a

change in long-term water storage in a glacierized watershed, and represents an additional flux of water that would not be present if the glacier system was in equilibrium with its environment (Radiç and Hock, 2010). These additional fluxes can affect the physical and chemical oceanography in coastal Alaska's bays and fjords (Reisdorph and Mathis, 2014).

Glacier cover changes in response to long-term changes in meteorological forcing and Beamer et al. (2017) have estimated future

hydrographs for the GOA in response to changes in precipitation, temperature, and glacier cover. They considered a variety of climate model outputs and representative concentration pathways (RCPs). For RCP 8.5, which corresponds to a scenario of comparatively high greenhouse gas emissions, they found that the overall runoff increased by 14%, but the runoff from glacier surfaces decreased by about 34%. Beamer et al. (2017) also found significant changes in the timing of the delivery of freshwater to the coast. In response to changes in temperature, precipitation, and glacier cover, summer flows were dramatically reduced, with

strong increases in autumn and winter flows. The overall (averaged over the GOA) hydrograph was estimated to change from one dominated by a summer a peak to one with two peaks; one due to spring snowmelt, the other due to autumn rains.

Glacier Bay (Figure 1b-c) is a fjord estuary in southeast Alaska that makes up part of Glacier Bay National Park and Preserve (GBNPP). The bay itself is roughly Y shaped, with maximum depths of approximately 500 m in the upper west and east arms, and

an overall volume of 162 km$^3$. In contrast, depths near the entrance sill are approximately 25 m. The tidal forcing of the bay is considerable, with a Great Diurnal Range (GT; difference between Mean Higher High Water (MHHW) and Mean Lower Low Water (MLLW)) of 3.36 m (data from NOAA Station 9452634; Elfin Bay, AK). The large tidal range produces strong tidal mixing that tends to destratify the water column. This effect counteracts the large freshwater input to the bay that tends to stabilize the water column. The result is a complex pattern of spatial and temporal variability of water column properties. Etherington et al.

(2007; their Figure 5) summarized 10 years of oceanographic measurements (CTD casts) made in Glacier Bay at a total of 24 stations. They aggregated their measurements by month and by region (West Arm, East Arm, Central Bay, and Lower Bay). The results showed that stratification was largest in the summer, due to the large runoff associated with ice-melt. Spatially, it was found that there was a strong up-bay gradient in stratification, with the weakest stratification found in the Lower Bay where shallow depths produced the strongest vertical mixing of the water column.


Etherington et al. (2007) correlated various water column properties (stratification, chlorophyll $a$, etc.) against physical variables such as day length, wind speed, air temperature, etc., in order to develop a better understanding of the ecology of the bay. While their discussion considered the role played by freshwater inputs, the lack of observational data (stream gauging) and hydrological modeling studies of Glacier Bay left their hypotheses untested. Hill et al. (2009) applied the regression equations for flow

exceedances (e.g., the discharge exceeded 50% of the time) and peak flows (e.g., the 10-year event) developed by the USGS



(Curran et al., 2003; Wiley and Curran, 2003) to Glacier Bay in order to help constrain the likely range of flows into the Bay. Their results suggested that the 10-year return interval discharge into the bay was approximately 10,000 $m^3$ $s^{-1}$ and that the 50% exceedance annual discharge was approximately 800 $m^3$ $s^{-1}$, however their study included a different contributing area, with watersheds on the southern side of Glacier Bay included.

As was the case with the GOA watershed as a whole, Glacier Bay is a region that continues to undergo dramatic change. Glaciers have retreated over 100 km since the end of the little ice age (LIA; Hall and Benson, 1995) and the volume of ice lost in the Glacier Bay region alone was enough to raise global sea levels by 0.8 cm (Larsen et al., 2005). This glacial retreat has led to rapid vegetation succession (Chapin et al., 1994) and to large uplift rates (30 mm $yr^{-1}$; Larsen et al., 2005) that produce falling relative sea levels.

The GRACE data for the Glacier Bay region show a downward trend of 12 cm $yr^{-1}$ w.eq., which is very close to the average decrease of 13.3 cm $yr^{-1}$ obtained for the entire GOA watershed (Luthcke et al., 2013; Beamer et al., 2016).

This paper presents the results of a hydrological modeling study of Glacier Bay. We understand it to be the first high-resolution (sub-km), process-based study of the water cycle in the region. Recall that the results of Hill et al. (2009) were statistical and only

provided a few representative flow values. Here, the goals are very different. We use an energy-balance model to evolve the snowpack and melt glacier ice after the seasonal snowpack has melted away. Also, our model results are output on a daily time step, which provide a variety of derived products (monthly averages, seasonal climatologies, etc.). Glacier Bay is a high-gradient landscape (rapid spatial changes in terrain, precipitation, e.g.) and we anticipate considerable spatial variability in both present hydrographs as well as future hydrographs. The results of this study will add to the understanding developed by Etherington et al.

(2007) and will provide constrained estimates of how much the coastal runoff in GBNPP will change in the future.

## 2 Study Area

The study area (Figure 2a) lies mostly within the boundary of GBNPP. Some watersheds originate outside the National Park boundary, and are located partially in Canada. The elevation ranges from sea level to heights in excess of 4500 m on Mt. Fairweather. The study area includes 3 individual watersheds, 4 grouped watersheds, and the fully aggregated GBNPP domain.

These various domains have been selected to illustrate the gradients in hydrologic inputs and outputs that exist in the region. See Table 1 for more details about the spatial extent, average elevations, and glacier coverage of each grouped and individual watershed.

The northern group of watersheds (North; Figure 2b) supplies freshwater to the mouth of Glacier Bay and constitutes the largest

sub-group in the study area (see Table 1). The western group of watersheds (West; Figure 2b) delivers freshwater to the Pacific Ocean directly. We further subdivide a portion of the North watershed into two smaller aggregated regions near the western (West-Arm; Figure 2c) and eastern (East-Arm; Figure 2c) regions of Glacier Bay. The two arms of Glacier Bay have notable differences in elevation, glacier cover, and water column properties, and the aggregated watersheds shown in Figure 2c correspond to similar regions investigated by Etherington et al. (2007) and a large portion of the domain from Hill et al. (2009).


Finally, we examine several individual watersheds within GBNPP. The first is a small group of watersheds that includes the Margerie and Grand Pacific tidewater glaciers terminating in the Tarr Inlet in the Western arm of Glacier Bay (Tarr; Figure 2d). The second is a highly glacierized region that includes Carroll Glacier, a land-terminating glacier with outlet lobes that deliver freshwater to the East-Arm and West-Arm of Glacier Bay (Carroll; Figure 2d). The last is a low-elevation, rain dominated





watershed in the Dundas River region that experiences occasional glacial-lake outburst floods from the adjacent Brady Icefield (Dundas; Figure 2d). We chose these three individual watersheds to illustrate and examine the various ice-melt, snowmelt, and rainfall dominated runoff patterns and the changes they may experience in future scenarios. The results of this study are categorized into the eight watersheds mentioned above. However, the focus of the results is on the aggregated GBNPP domain, and the

appendices contain the supplemental grouped and individual watershed results.

## 3 Data and Methods

### 3.1 Hydrologic Model

In this study we use a suite of models to simulate freshwater runoff to Glacier Bay for two climatological periods; 1979-2015 and 2070-2099. First, MicroMet (Liston and Elder, 2006a) is used to distribute the gridded reanalysis forcing data throughout the model

domain. Second, SnowModel (Liston and Elder, 2006b) is used to evolve the snowpack and melt glacier ice using energy-balance methods. This suite of models has been widely used in high latitude, highly glacierized environments including Alaska (Beamer et al., 2016; Beamer et al., 2017), the Arctic (Mernild et al., 2011; Liston and Hiemstra, 2011; Liston and Mernild, 2012; Mernild and Liston, 2012; Mernild et al., 2013; Mernild et al., 2014) and the Andes (Mernild et al., 2017a-d). Below we only briefly review the model components. Readers are directed to the source publications for full details on model algorithms and to Beamer et al.

(2016) for full details on the application to the GOA.

MicroMet (Liston and Elder, 2006a) is a meteorological distribution system for weather forcing datasets in high spatial resolution, distributed terrestrial modeling applications. The model relies upon the Barnes objective analysis scheme (Barnes, 1964; Barnes, 1973) for spatial interpolation of atmospheric variables, generating data fields at each time step and grid cell in the model domain

for eight atmospheric variables. The atmospheric variables required by MicroMet include surface level precipitation, wind speed and direction, relative humidity, and air temperature. Sub-models of MicroMet will calculate radiation fluxes if they are not available as inputs. Landcover and elevation datasets are also employed by MicroMet to establish relationships based on topographically and seasonally varying temperature lapse rates, and topography dependent wind and solar radiation fields.

SnowModel (Liston and Elder, 2006b) is a physically based model for estimating snowpack accumulation and ablation processes over a water year. Sub-models within SnowModel estimate the energy fluxes of the snowpack and generate the snow depths and snow water equivalence for each cell in the gridded domain. The primary input for SnowModel is the gridded forcing dataset of atmospheric conditions that vary throughout the simulation time period and get distributed throughout the model domain by MicroMet. SnowModel does have the ability to melt glacier ice after the annual snowpack has fully melted away, but it does not

include dynamic adjustments to the glacier cover volumes or extent. Therefore, SnowModel is able to simulate the hydrologic response of a fixed landscape, but it cannot model century-scale evolution of glacier cover.

### 3.2 Model Forcing Data

### 3.2.1 Elevation and Land Cover

The land surface elevation dataset is the NASA Shuttle Radar Topography Mission (SRTM; Farr et al., 2007) 90 m digital elevation

model (DEM) resampled to 250 m. A model grid resolution of 250 m was selected for the present study as a compromise between desired high spatial resolution and the accompanying computational demands. The 250 m North American Land Cover Monitoring



System 2010 (NALCMS) dataset was used for the land cover characterization. In order to obtain the most recent data on glacier coverage we used the Randolph Glacier Inventory (RGI; v.3.2; Pfeffer et al., 2014).

### 3.2.2 Historic Climate Data

5   The Modern-Era Retrospective Analysis for Research and Applications (MERRA) weather reanalysis product from NASA's Global Modeling and Assimilation office was chosen as the forcing meteorologic dataset for SnowModel during the simulation period. MERRA uses a data assimilation method for conventional observations of atmospheric data from irregularly spaced weather stations from around the world, collected by the National Climatic Data Center (Rienecker et al., 2011). The MERRA data are available at a nominal spatial resolution of 67 km and a temporal resolution of 3 hr. Variables available from the MERRA dataset include precipitation, 2 m air temperature and relative humidity, and 10 m wind speed and direction.

10  ### 3.2.3 Historic Evapotranspiration Data

Beamer et al. (2016) developed a soil moisture and evapotranspiration sub-model for the MicroMet and SnowModel framework. They demonstrated good agreement with Moderate Resolution Imaging Spectroradiometer (MODIS) satellite estimates of evapotranspiration (ET). For this study, MODIS-based ET values are calculated from the MOD16A2 8-day, 1 km resolution product. Monthly and annual climatologies based on averages from January 2000 through December 2014 are derived for each of 15  the eight grouped and individual watersheds. These monthly MODIS-based ET values are plotted on the historic runoff figures but not calculated as a loss in the water balance because the ET values are derived separately from the modeling process.

### 3.3 Oceanographic Data

Standard oceanographic conditions for GBNPP are taken from a long-term (1993-present) observational dataset created by the U.S. National Park Service's Southeast Alaska Inventory & Monitoring Network (SEAN). The SEAN dataset includes depth 20  profiles of water column properties, including temperature and salinity, from Conductivity-Temperature-Depth (CTD) sensor casts at each of twenty-two active stations (Figure 2). As of the sampling protocol imposed in 2014, all stations are sampled in midsummer (July) and midwinter (Dec), and a subset of eight stations are also sampled monthly from March through October to capture the rapid temporal variability of the spring-summer season (Johnson and Sharman, 2014). Prior to 2014, stations were sampled between four and nine times per year, at various months, providing sufficient sampling data to calculate long-term monthly 25  averaged conditions. The CTD station locations are spaced throughout GBNPP approximately five nautical miles apart. The vertical resolution of the CTD casts is approximately one meter.

### 3.3.1 Oceanographic Data Analysis

Well-defined isohalines present in the oceanographic dataset allow for point estimates of freshwater content (FWC) at station locations within GBNPP (McPhee et al., 2009). FWC can be calculated as the depth-integrated freshwater anomaly relative to a 30  defined reference salinity, following McPhee et al. (2009) and earlier work by Carmack et al. (2008):

$$FWC = \int_{z_{lim}}^{0} (1 - S(z)/S_{ref}) dz \tag{1}$$

where S(z) is the depth-dependent salinity (practical salinity scale, unitless) and FWC has dimensions of length. The lower limit 35  of integration $z_{lim}$ is taken to be the bathymetric depth at each station. At the lower limit, several casts appear to have terminated



after reaching depth-invariant salinity readings, rather than reaching the bathymetric depth. For these casts, the final recorded salinity was used to extend the salinity profile to $z_{lim}$. Missing data at the upper limit of the profile were filled using spline interpolation; for data gaps exceeding 5 m from the surface, the cast was ignored.

5     Representative of highly saline inflowing waters of the GOA, $S_{ref}$ was chosen as an upper-end reference salinity of 34.8 practical salinity (Carmack et al., 2008). In this analysis, choice of $S_{ref}$ was found to have no significant influence on seasonal changes in FWC at a given location. FWC values at individual stations were then interpolated to the entire bay surface and spatially integrated, allowing for the calculation of a freshwater volume (FWV). This interpolation was done using a splines with tension method (Wessel and Bercovici, 1997).

10    **3.4 Model Calibration**

Recent studies (Beamer et al., 2016; Lader et al., 2016) have investigated the accuracy and biases of the MERRA reanalysis product in coastal Alaska compared to other reanalysis products such as ERA-Interim (Dee et al., 2011), CFSR (Saha et al., 2010), NCEP-NCAR (Kalnay et al, 1996), NARR (Mesinger et al., 2006), and others. For this study, we rely on the model calibration of Beamer et al. (2016; their section 3.4). For each of 4 reanalysis products, they calibrated model parameters based on observations of 15    streamflow (Q) and glacier mass balance (B). While Beamer et al. (2016) identified the CFSR product as the 'best overall' for the GOA region, they found that MERRA was superior at the Mendenhall Glacier observational station, which is the closest calibration point (< 25 km) to GBNPP. The Coefficient of Determination ($r^2$) and Nash Sutcliffe Efficiency (NSE) scores for the MERRA product for B and Q were 0.80 and 0.95 for $r^2$, respectively, and 0.67 and 0.91 for NSE, respectively.

**3.5 Model Forecast Datasets**

20    **3.5.1 Future Climate**

Local to regional scale studies of future runoff are complicated by the fact that future climate model outputs are typically produced at a spatial resolution of $1 - 2°$. Beamer et al. (2017) dealt with this by using high-resolution (2 km) historic and future climatologies (30-year averages available for each month of the year) to perturb the historic weather reanalysis datasets. This 'delta' or 'scaling' method of constructing future weather datasets is widely used in climate change studies (see Fowler et al., 2007 for a review). 25    While it has the disadvantage of not capturing future changes in the frequency distributions of weather variables, if the primary interest is in monthly averages or climatologies, then this deficiency is of little consequence. Beamer et al. (2017) used the climatologies from the Scenarios Network for Alaska Planning (SNAP) project which are based upon CMIP5 climate scenarios. SNAP has results for the five best performing (for Alaska; see Table 2) climate models as well as a result representing the mean of the 5-model ensemble. Although future climate simulations exist for numerous RCP (representative concentration pathway) 30    scenarios, in this study we restrict ourselves to the RCP 8.5 scenario and to the 5-model mean.

**3.5.2 Future Glacier Cover**

Since SnowModel does not model glacier dynamics (i.e., glacier advance and retreat), the historic and forecast simulations represent the response of a particular landscape to the climate. For the historic run, the landscape represents the RGI 2014 glacier extent. For the future run, the glacier mask is adjusted as described in Beamer et al. (2017). Essentially, the glacier cover is adjusted, 35    using the accumulation area ratio (AAR) method of Paul et al. (2007), under the assumption that glaciers will be in equilibrium with climatic conditions. We note that there are modeling efforts that attempt to directly model ice flow dynamics (e.g., Clarke et





al., 2015; Ziemen et al., 2016) but those efforts come with significant input data requirements. Our approach can be thought of as a leading-order test of the sensitivity of the hydrologic scale to plausible landscape changes.

To evolve the glacier extent using the AAR method of Paul et al. (2007), two key parameters are required. The first is the value of

the AAR, which is the ratio of the accumulation area of a glacier to the total area of the glacier. The second is the change in the equilibrium line altitude (ELA) of the glacier, due to changing climatic conditions. The steady-state AAR ($AAR_0$) was chosen to be 0.65, based on the observations of several benchmark Alaskan glaciers by Mernild et al. (2013). While some studies have suggested that $AAR_0$ values change in the future, Beamer et al. (2017) found that the assumption of a steady-state AAR (i.e., keeping AAR fixed at 0.65 for the future runs) provided estimates of future glacier changes that are in accord with other published

values (Huss and Hock, 2015; their figure S10; McGrath et al., 2017). As a result, we similarly assume $AAR_0$ to be equal to 0.65 for the future runs. Regarding the ELA, we use the results of Huss and Hock (2015, their figure S9) and assume an ELA increase of 400 m for the RCP 8.5 scenario, based on their modeled ELA changes between 2010 and 2100.

### 3.5.3 Future Climatologies

The MERRA reanalysis data were used with the model configuration described above to produce a 30-year historic simulation of

runoff. The daily output was temporally aggregated to monthly values and then climatologies were produced for each month of the year. The future runoff estimates were obtained using the coarser (1-km) model results of Beamer et al. (2017) and a scaling method similar to that described in Section 3.4.1 in the context of meteorological variables. Scaling methods are rooted in the idea of a separation of scales. A certain variable, say precipitation, may have a high degree of spatial variability, but *changes* in this variable (from historic to future conditions) have a much lower degree of spatial variability. In this way, climatologies from coarse

(degree scale) climate model output can be used to create anomaly fields that may be recombined with high-resolution historic results to create high-resolution future projections. In the context of runoff, the Beamer et al. (2017) 1 km historic and future results are used to create runoff scaling factors per watershed that are applied to the higher resolution (250 m) historic runoff results created for Glacier Bay in this study. At the end of this process, we have both historic and future climatologies of runoff per watershed that allow us to quantify seasonal changes in the delivery of freshwater to Glacier Bay.

**4 Results**

The following results for changes in temperature, precipitation, SWE, and runoff are based on the 35 year historic climatologies from the MERRA-forced, 250 m model output. The 30 year forecast climatologies are based on Beamer et al. (2017), which is CFSR-forced, 1 km model output derived from the scaling factors discussed previously in section 3.4. The historic and forecast results are spatially aggregated by watershed and discussed below.

**4.1 Changes in Temperature**

The changes in watershed average temperature from the historic to forecast scenario reveal the most substantial temperature increases occur from October to December, followed by May to July, for the aggregated GBNPP watersheds (Figure 3a). These temperature changes are characterized in the figure following Eq. (2):

$$\Delta_{TEMP(C)} = \text{Temp}_{i,k}^{fore} - \text{Temp}_{i,k}^{hist} \tag{2}$$

where $i$ is the month, $k$ is the watershed, *Temp* is the climatological average temperature (C), *fore* is the forecast scenario, and *hist* is the historical scenario. As a result of the model runs, all months in all watersheds experience a temperature change greater than




degrees (C) due to the choice of the RCP8.5 scenario for the forecast scenario. This is likely amplified by the high elevation gradients in GBNPP topography and the high latitude environment that create temperature changes of more than 4 degrees (C) for many of the watersheds in multiple months (Figure 3a). The historic average winter (DJF) temperature in GBNPP is -4.1 degrees (C), while the forecast average DJF temperature is only slightly below zero, at -0.2 degrees (C). These changes in average seasonal,

monthly, and annual temperatures are driving many of the changes in the modeled precipitation, snowfall vs. rainfall partitioning, snowpack evolution and ablation, glacier ELA and AAR, and the seasonality of the modeled runoff climatologies.

### 4.2 Changes in Precipitation

The changes in precipitation in GBNPP from the historic to forecast scenario can be divided into three categories: changes in total precipitation, changes in monthly partitioning of rainfall vs snowfall, and changes in the ratio of snowfall w.eq. to total precipitation

(SFE/P). First, the changes in total precipitation include increases in precipitation in GBNPP from the historic average of 3.40 m yr$^{-1}$ to a forecast average of 3.71 m yr$^{-1}$, which represents a 9.0% average annual increase in precipitation. These average total precipitation changes (%) include variability among watersheds and between seasons, with October and November containing the largest increases in precipitation $\Delta$ values, and January containing the largest decreases in precipitation $\Delta$ values (Figure 3b). These precipitation changes are characterized in the figure following Eq. (3):

$$\Delta_{PREC(\%)} = \left( \frac{Prec_{i,k}^{fore} - Prec_{i,k}^{hist}}{Prec_{i,k}^{hist}} \right) x\ 100 \qquad\qquad (3)$$

where $i$ is the month, $k$ is the watershed, *Prec* is the climatological average precipitation (m) value, *fore* is the forecast scenario, and *hist* is the historical scenario.

We use a common metric to characterize annual and monthly change in snowfall from the historic to forecast simulations: the

snowfall w.eq. (SFE) to total precipitation (P) ratio (SFE/P; Mote, 2003; Mote, 2005; Knowles et al., 2006; Zhang et al., 2000). The SFE/P metric can illuminate the snowfall trends within a region, where 1 represents all precipitation falling as snow and 0 represents no snowfall in the watershed over the time period of interest. Changes in SFE/P are calculated in the figure following Eq. (4):

$$\Delta_{SFE/P} = \left( \frac{SFE_{i,k}}{Prec_{i,k}} \right)^{fore} - \left( \frac{SFE_{i,k}}{Prec_{i,k}} \right)^{hist} \qquad\qquad (4)$$

where $i$ is the month, $k$ is the watershed, *Prec* is the climatological average precipitation (m), *SFE* is the climatological average snowfall equivalent (m), *fore* is the forecast scenario, and *hist* is the historical scenario. Characterized this way, when $\Delta_{SFE/P}$ is negative it means more precipitation is falling as rain in the forecast and when $\Delta_{SFE/P}$ is positive, more precipitation is falling as snow. All eight watersheds experience negative annual $\Delta_{SFE/P}$ from the historic to the forecast model runs, even though annual changes in precipitation are primarily increasing from the historic to forecast scenario (Figure 3c). The highest and lowest mean

elevation watersheds, Tarr and Dundas, respectively, display an opposite behavior in the magnitude of their seasonal SFE/P values and this relationship will be further investigated in the discussion section. These results are congruent: the changes in temperature, changes in total annual precipitation, changes in snowfall vs. rainfall partitioning, and changes in SFE/P all point towards a landscape that is less dominated by snowfall and is increasingly influenced by rainfall in the forecast scenario.

To supplement the $\Delta_{SFE/P}$ analysis, the results of the historic and forecast precipitation are analyzed in terms of monthly snowfall vs. rainfall partitioning for each watershed. While this type of precipitation partitioning may be a relatively crude characterization



of a complex atmospheric system, where mixed snowfall and rainfall occur simultaneously, this distinction is practical and appropriate for our research questions and the application of SnowModel. For the purposes of this paper, the dominant process is simply the one that is ≥50% of total precipitation. The precipitation partitioning results for GBNPP (Figure 4) display an annual average that shifts from snowfall dominated precipitation historically (58.2% snowfall vs. 41.8% rainfall) to a rainfall dominated precipitation regime in the forecast scenario (24.1% snowfall vs. 75.9% rainfall). Additional historic and forecast precipitation climatologies can be found in *Appendix A* for each of the eight grouped and individual watersheds. In summary, the low-elevation Dundas watershed is the only rainfall dominated watershed in the historic model runs, while all other watersheds are snowfall dominated. In contrast, only the highly glacierized and high-elevation Tarr and Carroll watersheds remain snowfall dominated in the forecast scenario. All others switch to rainfall as the primary annual precipitation process.

**4.3 Changes in Glacier Coverage**

The glacier change map (Figure 5) is the result of adjusting the RGI 2014 glacier outlines in GBNPP by increasing the ELA by +400 m, while keeping the AAR fixed at 0.65 according to Huss and Hock (2015). This change map displays the static glacier cover used for the historic runs as well as the static glacier cover used for the forecast runs. Recall that SnowModel does not dynamically adjust glacier extent, so these glacier changes represent two distinct landscapes that remain in equilibrium with their environment for the duration of the modeled time period. In the aggregated GBNPP watersheds, the forecast scenario contains a 58.8% decrease in glacier cover compared with the RGI 2014 map, from a total historic surface area of 4092 km$^2$ to 1687 km$^2$ in the forecast scenario.

**4.4 Changes in Snow Water Equivalent**

Snow water equivalence (SWE) was modeled for the historic and forecast scenarios and aggregated for all GBNPP watersheds by mean monthly depth (Figure 6). Peak SWE historically occurs in April and while the timing remains unchanged in the forecast scenario, GBNPP watersheds lose 46% of mean peak SWE in the forecast scenario. The monthly relative changes in SWE from the historic to forecast scenario range from -44% in March to -70% in September. These loses are in line with Shi and Wang's (2015) investigation into Northern Hemisphere changes in SWE based on the RCP 8.5 scenario (their Figures 4c & 6f). The magnitude of the SWE losses in the forecast scenario will directly affect the timing and volume of runoff generated from snowmelt.

**4.5 Changes in Runoff**

The historic runoff hydrograph for GBNPP is partitioned into the components of ice-melt, snow melt, and rain runoff, and includes the MODIS-based ET values (Figure 7a). Each component is given in mean monthly depth (m) instead of total volume (km$^3$). This normalization by watershed area enables straightforward comparison between individual and grouped watersheds. The historic and forecast volumes can be found in Table 3. Ice-melt is runoff generated when glacial ice is melted after the seasonal snow disappears in each glacier grid cell. This definition of ice-melt, as a runoff component, does not necessarily represent glacier volume loss, due again to the fact that SnowModel does not dynamically change glacier extent or volumes. These runoff component values for a watershed of interest are calculated by summing the values for all model grid cells in each watershed. Unlike the work of Beamer et al. (2016; 2017) we did not route the runoff across the landscape to the coastline. In GBNPP, the average distance from a grid cell to its coastal outlet is only 9.0 km. Given this short distance, and the fact that our interest here is in seasonal climatologies of runoff, and not daily time series, this is a justifiable simplification.



The historic runoff hydrograph for GBNPP displays low runoff quantities during the winter months, with snowmelt dominating spring and early summer, ice-melt supplementing runoff in mid-summer, and rain runoff dominating early fall (Figure 7a). The average MODIS-based ET loss for GBNPP is 0.28 m yr$^{-1}$, while the average historic precipitation is 3.40 m yr$^{-1}$, which makes ET loss <9% of precipitation on average for all watersheds. The forecast scenario runoff total for GBNPP is 3.96 m yr$^1$, and displays

a distinct flattening of the annual runoff hydrograph in terms of quantity and timing of snowmelt, as well as a decrease in ice-melt to Glacier Bay (Figure 7b). The historic and forecast runoff hydrographs for each grouped and individual watershed can be found in Appendix B, and Figure 8 presents changes in the runoff components in the forecast scenario. In many of the watersheds in the GBNPP domain, there is an overall annual increase of runoff volumes in the forecast simulations, with much of that increase sourced from changes in rain runoff. These increases in rain runoff originate from higher temperatures in the forecast scenario,

losses in glacier area, increases in overall precipitation, and increases in the rainfall component of precipitation.

**4.6 Historical Freshwater in Glacier Bay**

A climatology of the month-to-month changes in FWV (ΔFWV) for various sub-regions of Glacier Bay is shown in Figure 9. The trends are similar for all three regions. Positive values of ΔFWV are observed in summer months when the strong runoff fluxes from snow and ice-melt outpace the ability of water to flush out through the bay mouth. Negative values are observed in winter

months when runoff is low and the bay is able to flush out the accumulated freshwater. The larger values in the West Arm (vs. the East Arm) are due to the larger watershed area, higher mean elevation, and greater glacier cover.

Long-term changes in July FWC were also examined. Note that July was chosen since that month had the most measurements throughout the bay. Spatially averaged July FWC, or $\overline{FWC}$, for various bay sub-regions, was found by interpolating July FWC

observations across GBNPP and then averaging across each bay sub-region (Figure 10). The linear trend lines shown in Figure 10 indicate little significant change in $\overline{FWC}$ over the study period (1993-2017). The West-Arm July $\overline{FWC}$ observations are the exception, increasing with a rate of 8.3 cm per year (p-value of 0.109).

**5 Discussion**

The distinct changes observed in the study area watersheds motivate investigation into the controlling physical characteristics of

the various landscapes within GBNPP. For example, in Figure 3c, the patterns of Δ$_{SFE/P}$ in the Tarr and Dundas watersheds have opposing seasonal trends. The Tarr watershed has a comparatively high mean elevation and sees only small magnitudes of winter Δ$_{SFE/P}$. This may be because much of the watershed remains above the snow line in the forecast run (see Figure 11a). Tarr also displays high magnitudes of summer Δ$_{SFE/P}$, and it has historically been one of the few watersheds that receives significant snowfall precipitation throughout the summer, again due to the high elevation. As a point of contrast, the Dundas watershed has the lowest

mean elevation of the eight study watersheds (see Figure 11a). It sees high magnitudes of Δ$_{SFE/P}$ in the winter, but very small magnitudes in the summer. The former is attributable to the future snowline in Dundas increasing above the maximum watershed elevation and the latter is due to very small amounts of historic summer snowfall precipitation in Dundas. This initial comparison between the changes in Tarr and Dundas suggests a need to further investigate landscape dependencies and the seasonal aspect of snow precipitation, especially with elevation.


Snow distribution and elevation in mountain environments is highly correlated (Bales et al., 2006; Fassnacht et al., 2003; Welch et al., 2013) and in maritime regions, understanding the role of elevation distributions within a watershed is important in the context



of changing climate and the snow-to-rain transition zone (Jefferson 2011). Histograms of elevation, along with polar coordinate plots of slope and aspect for GBNPP, Dundas, and Tarr are given in Figure 11 to help illuminate the relationships between elevation, temperature and precipitation change, and process change. Recall that these changes take the form of negative $\Delta_{SFE/P}$ values in all watersheds (Figure 3c). When considered in relation to the monthly or seasonal average freezing-line altitude (FLA),

the magnitudes of the Dundas and Tarr seasonal SFE/P changes (Figure 3c) begin to make sense. We calculated the historic and forecast FLAs by averaging the winter and summer temperatures across all historic years (1979-2015) and all forecast years (2070-2099) and extracted the elevational bands that corresponded with the 0 °C or rain-to-snow transition line. For this analysis, the most important aspect may be the proportion of the watershed area located between the historic seasonal FLA and forecast seasonal FLA. In Dundas, the winter FLA increase of several hundred meters in the forecast scenario means that a large proportion of the

watershed would receive rainfall when it previously received snowfall. In contrast, when the Tarr watershed is subjected to the same several hundred meter winter FLA increase, only a small proportion of the watershed is affected by that increase (see Figure 9a), thus undergoing lower magnitudes of $\Delta_{SFE/P}$ than Dundas. The summer FLA increase of >1000m means that Dundas will likely receive insignificant summer snowfall in the forecast scenario, but Tarr will experience higher magnitudes of $\Delta_{SFE/P}$. This is because a large proportion of the Tarr watershed lies between the historic and forecast summer FLAs, but Dundas always received

insignificant snowfall historically during the summer.

Similarly, the distribution of snowpack on the land surface has landscape dependencies on aspect and slope. Regarding topographic slope, Tarr has proportionally more steep slopes than GBNPP and Dundas, and steep slopes tend to accumulate snow in the same locations year after year by way of sloughing, avalanching, and wind drift, distributing snow to the lesser inclined accumulation

areas (Figure 9b; Bloschl and Kirnbauer, 1992; Grunewald el al., 2010; Grunewald et al., 2014). The average aspect in Tarr is dominated by the northeastern direction, which increases shading and creates more oblique angles of incoming solar radiation, which affects snow water equivalent distribution and timing of meltwater. Alternately, the average aspect in GBNPP and Dundas is South to Southwestern and these aspects receive more direct incoming solar radiation angles and will affect accumulation patterns and meltwater patterns differently in these watersheds (Elder et al., 1989, Marks et al., 1999). This study acknowledges

these landscape dependencies and we attempt to briefly characterize some of them as controls on the modeled processes. However, further characterization of the landscape spatial gradients and controls is beyond the scope of this study, while higher resolution observations and modeling will be necessary to better understand their effects on runoff processes in the future.

This examination of the source components of runoff to Glacier Bay is partially limited by a lack of long-term validation datasets

for stream flow and other long-term weather station forcing datasets within the GBNPP model domain. However, an effort to parameterize and calibrate SnowModel based on the results of other recent, larger-scale modeling projects was made, as previously noted in Section 3.3 according to Beamer et al. (2016; 2017). While implementation of SnowModel using additional validation and forcing datasets would likely improve the accuracy of the results, no regional stream flow, SWE observations, or weather station datasets exist at the appropriate locations or scales. This highlights the need for multiple types of monitoring systems to be

implemented in GBNPP in order to decipher future changes in glaciers, snowpack and precipitation, and runoff processes in GBNPP. Additionally, other important fluxes were not characterized in this study due to decisions in the model chain process, most notably snow density characterization which allows for rain on snow (ROS) events to be examined. For this project, when rain precipitation occurs on top of an existing snowpack, ROS was characterized simply as increasing the snow water equivalent in the snowpack. Even though it is known that ROS runoff events generally occur at snow densities greater than ~550 kg m$^{-3}$, the

final results do not describe the volume or frequency of ROS events since the snow density output was not necessary or desirable




for our research interests. However, the results of this study reveal a shift from snowmelt dominated runoff historically to rain runoff in the forecast scenarios, and understanding the timing and spatial extent of ROS events may prove to be an important area of research in the future.

We include the historical freshwater analysis of Glacier Bay because long-term meteorological datasets or streamflow datasets do not exist for the study area. The inclusion of the observational CTD dataset allows the modeling effort to be contextualized. The most notable is the monthly timing of the historical runoff in GBNPP (Figure 7a) as it relates to the monthly fluctuations of freshwater volumes from the CTD analysis (Figure 9). Not only is the runoff timing confirmed by the observations, but the relative magnitude of the proportion of freshwater originating from the West-Arm and East-Arm watersheds is also confirmed by the

observations (Figure 7a; Figure 9). Since the modeled runoff volumes for the forecast scenario (Figure 7b; Appendix B) exhibit differences in timing and magnitude from the historical model runs (Figure 7a; Appendix B), we can assume that the influx of freshwater from the land surface to Glacier Bay in the forecast scenario will reflect those changes in timing and magnitude. From the historical model runs, July is the month with the most combined runoff from the various freshwater sources. Importantly, the trends in July freshwater content from the observations (Figure 10) display a non-stationary system from 1993 to the present. The

modeled changes in timing and magnitude of runoff from the land surface into Glacier Bay will have effects on bay ecology in the future and these observations show that the system is already in flux.

A key source of uncertainty in the present study is the determination of the future glacier cover. We relied on the findings of Beamer et al. (2017) to guide assumptions of future ELA increases and AAR changes, if any. Their decisions were, in turn, based

on regional-scale (Alaska-wide) modeling studies of glacier change (Huss and Hock, 2015) and on decadal-scale observational studies of glacier mass balance based on altimetry (Larsen et al., 2015) and gravimetry (Arendt et al., 2008). Our results for GBNPP show a change of -58% in glacier covered area. Huss and Hock (2015) give a figure of -32% for change in glacier volume in all of Alaska. It is difficult to directly compare these two, given that they are for different domains (local vs. regional) and for different variables (area vs. volume). To our knowledge, local-scale modeling studies of glacier change in just GBNPP are not available.

We note the work of Alifu et al. (2016) who use a variety of remote sensing products to quantify observational changes in mean snow line altitude (MSLA) and mean snow accumulation area ratio (MAAR) in GBNPP during 2000-2012. Their results support the general trends of the present study, in terms of reductions in area change and increases in MSLA, but the duration of their study is quite short in comparison to the century-scale nature investigated in the present study.

**6 Conclusions**

In this study, a high spatial resolution, distributed snow evolution and ablation model, SnowModel, is used to estimate current and future freshwater runoff into Glacier Bay, Alaska. The model is forced using the MERRA weather reanalysis product to create 35 year climatologies of precipitation, temperature, and the source components of runoff, including rainfall, snowmelt, and ice-melt. The future scenario applies the SNAP temperature and precipitation anomalies from the mean of 5 climate models for the years 2070-2099 based on the RCP 8.5 emissions scenario. The physical complexity and variability of the region produces a wide variety

of historic and forecast hydrographs within GBNPP, including rainfall, snowmelt, and ice-melt dominated responses depending on the season and watershed. The timing and relative scaling of the historical inputs of freshwater from the study area watersheds are validated by a long-term oceanographic dataset from the Southeast Alaska Inventory and Monitoring Network in Glacier Bay. The mean annual runoff to Glacier Bay in the forecast scenario will increase by 12.2% from the historic average, with much of the increased runoff sourced from rain inputs. The peak flows to the Glacier Bay fjord estuary will shift from late-summer to early-



fall, and the effects of these profound changes in freshwater runoff timing will be experienced across the estuarine environment and biological communities.

## 7 Author Contributions

Ryan Crumley and David Hill designed the research questions, chose the methods, and the oversaw the analysis of the results for the manuscript. Jordan Beamer and Ryan Crumley developed the scripts that were applied to the model output and performed the model simulations. Elizabeth Holzenthal contributed the oceanographic data analysis. Ryan Crumley prepared the manuscript with contributions from all of the co-authors.

## 8 Competing Interests

The authors declare that they have no conflict of interests.

## 9 Acknowledgements

Funding for this study was provided by the National Park Service and Glacier Bay National Park and Preserve under Pacific Northwest CESU Agreement #HBW07110001 and Federal Grant #P15AC01065.

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





**Figures**

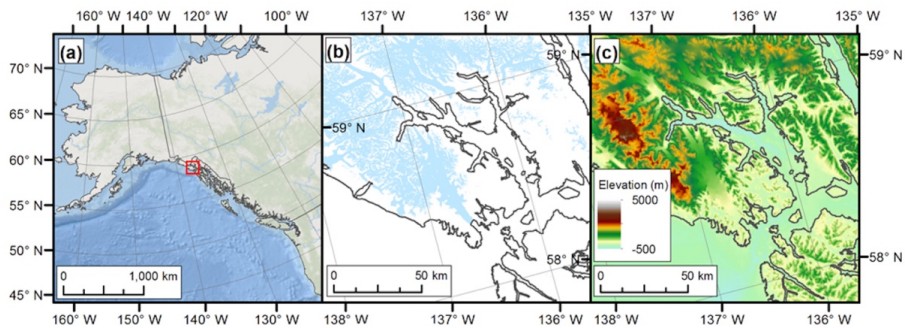

Figure 1: Study Area Map: (a) – Overview of northern Gulf of Alaska; red box shows extent of panels (b) and (c). (b) – Glacier cover (blue) in the Glacier Bay region from the Randolph Glacier Inventory (RGI; Pfeffer et al., 2014). (c) – Bathymetry and elevation in the Glacier Bay region from the Southern Alaska Coastal Relief Model (Lim et al., 2011).



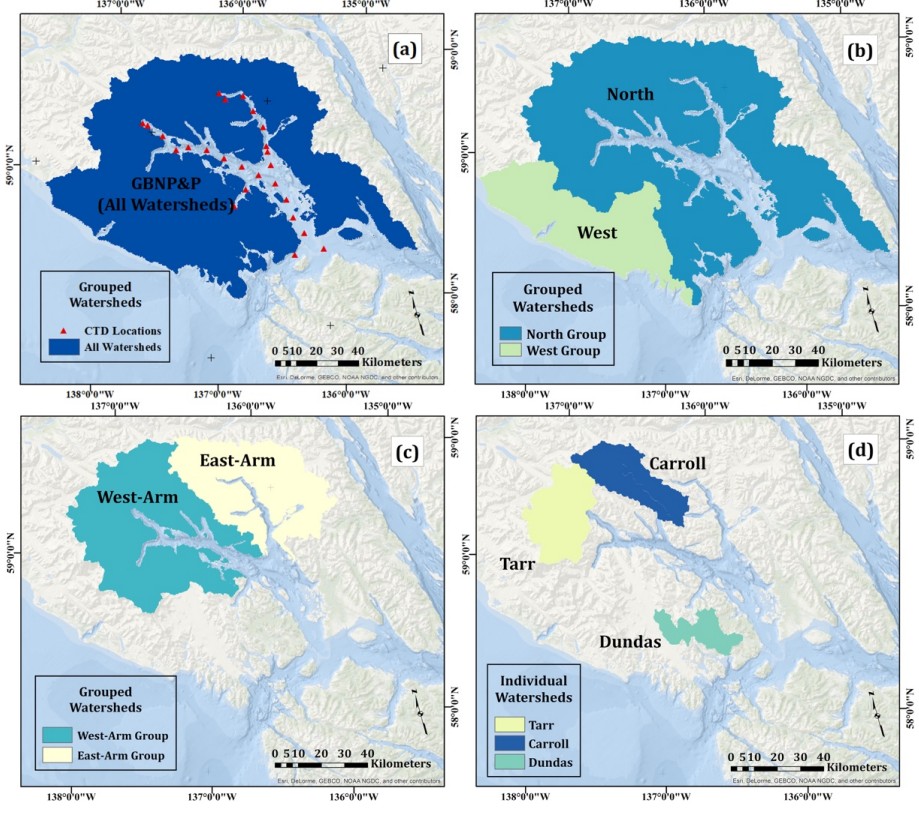

**Figure 2: Watershed Maps: (a) – All watersheds in the GBNPP group and the locations of the CTD casts (discussed in 3.3). (b) – North and West grouped watersheds. The North delivers freshwater to the main stem of Glacier Bay, and the West delivers water to the Pacific Ocean. (c) – The upper-bay grouped watersheds that deliver freshwater to the East-Arm and West-Arm of Glacier Bay. (d) – The 3 individual watersheds: Tarr, Carroll, and Dundas.**





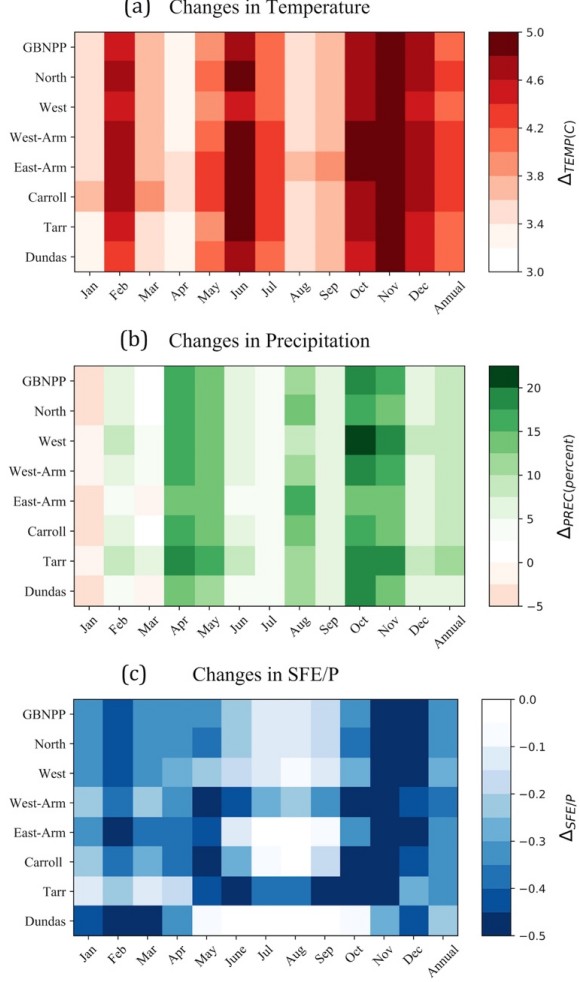

**Figure 3: Temperature and Precipitation Changes: (a) – Monthly and annual temperature changes (°C) from historic (1979-2015) values by watershed, based on temperature anomalies from the RCP8.5 scenario (2070-2099). (b) – Monthly and annual precipitation changes (%) from historic (1979-2015) values by watershed, based on the RCP8.5 scenario (2070-2099). (c) – Monthly and annual snowfall water equivalent to precipitation (SFE/P; unitless) changes from historic (1979-2015) values by watershed, based on the RCP8.5 scenario (2070-2099).**





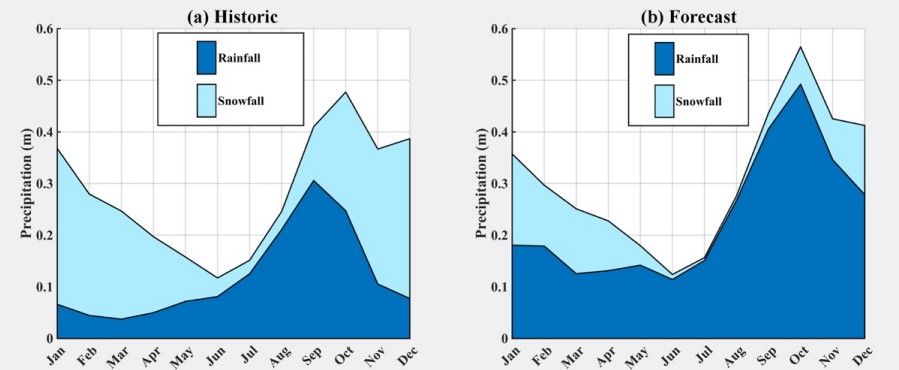

**Figure 4: Precipitation Climatologies: (a) – The domain aggregated GBNPP historic (1979-2015) precipitation climatology, partitioned into snowfall and rainfall constituents. (b) – The domain averaged and aggregated GBNPP forecast (2070-2099) precipitation climatology, partitioned into snowfall and rainfall constituents.**

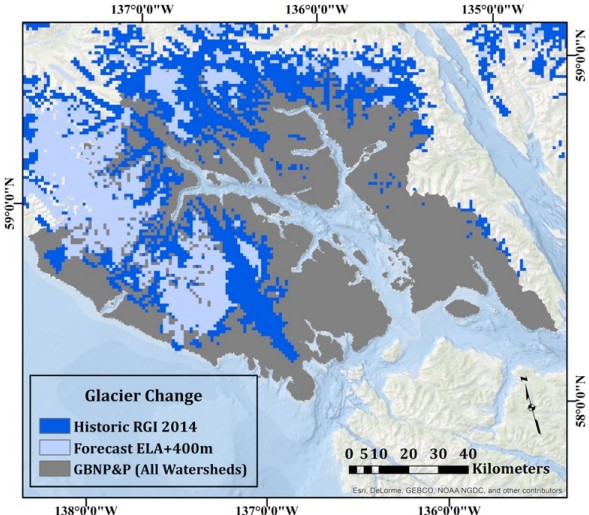

**Figure 5: Glacier Change Map: Changes in glacier extent in the study area based on the Randolph Glacier Inventory (RGI) 2014 glacier locations for the historic scenario (1979-2015) and the +400 m change in equilibrium line altitude for the forecast scenario (2070-2099) using the RCP 8.5 scenario.**



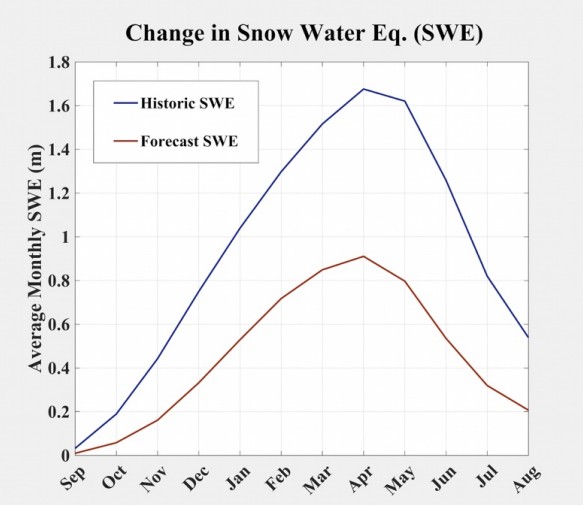

**Figure 6: Monthly snow water equivalence (m) averaged for the entire GBNPP domain for both the historic (1979-2015) and forecast (RCP 8.5 scenario; 2070-2099) scenarios.**

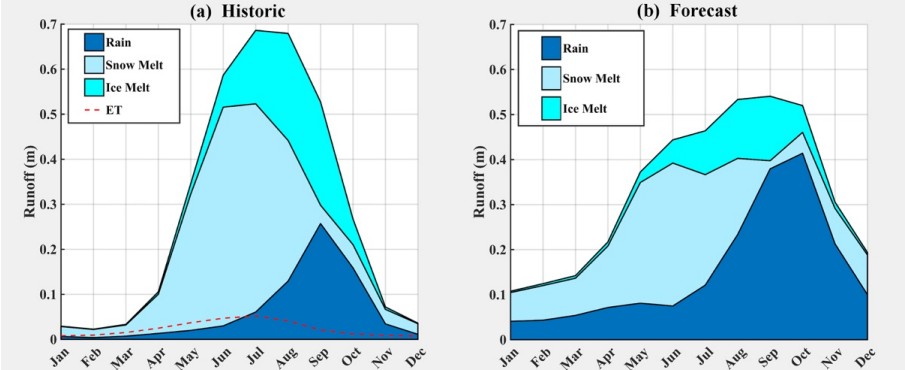

**Figure 7: Runoff Climatologies: (a) – The domain averaged and aggregated, historic (1979-2015) GBNPP runoff climatology partitioned into the constituents of snowmelt, ice-melt, and rain runoff. The historic (2001-2014) MODIS-based evapotranspiration estimates are**
10 **included on the historic plots, but the amounts are not subtracted from the modeled for runoff climatology because they were derived separately from the modeling process. (b) – The domain averaged and aggregated GBNPP forecast (RCP 8.5 scenario; 2070-2099) runoff climatology. Appendix B contains the historic and forecast runoff climatologies for each of the eight grouped and individual watersheds in the study area.**



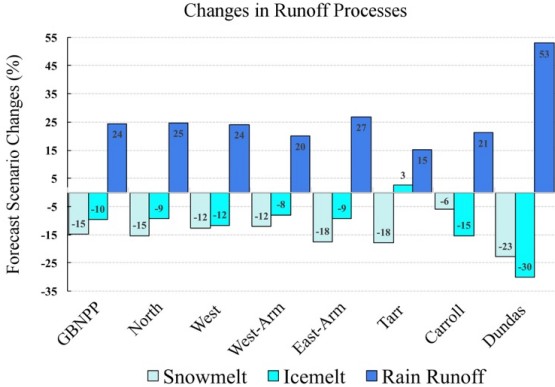

**Figure 8: Runoff process change by watershed in the forecast scenario (RCP 8.5 scenario; 2070-2099), partitioned into snowmelt, icemelt, and rain runoff.**

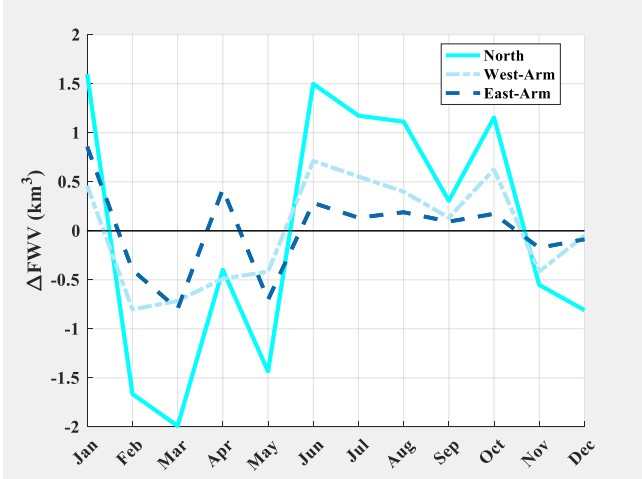

**Figure 9: Month to month changes in freshwater volume (ΔFWV) for the historical period of record (1993 to present) for various subregions (see Figure 2) of Glacier Bay.**



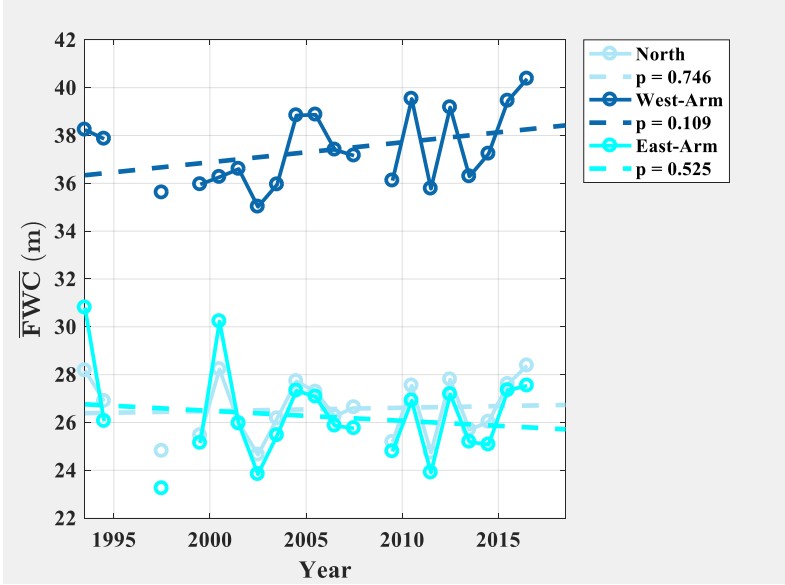

**Figure 10: Spatially averaged July freshwater content ($\overline{FWC}$) from 1993 to present, interpolated from the 22 active CTD locations within Glacier Bay (see Figure 2). Linear trends are indicated by dashed lines.**





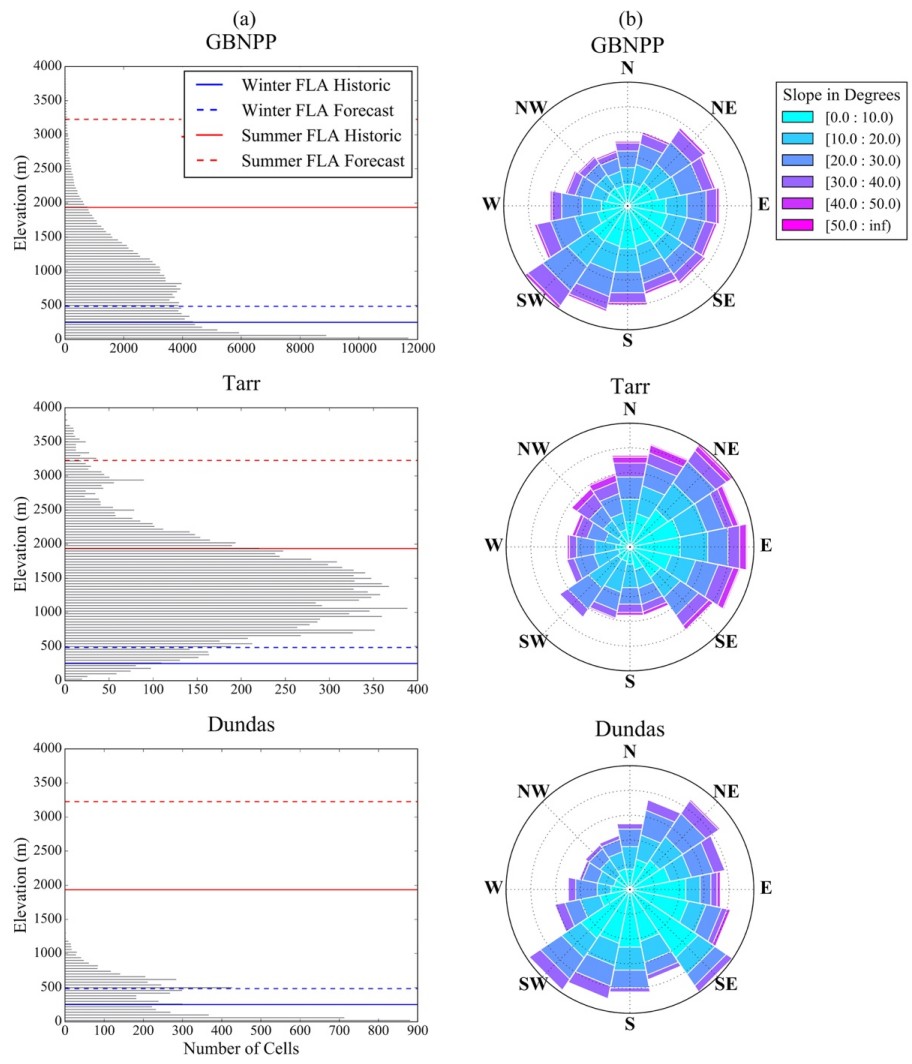

Figure 11: Landscape Characteristics: (a) – Elevation histograms for GBNPP, Tarr, and Dundas watersheds with the average winter and summer freezing line altitudes (FLA) plotted in blue and red, respectively. Historic scenario (1979-2015) lines are solid and forecast scenario (RCP 8.5; 2070-2099) are dashed. (b) – Polar coordinate plots for GBNPP, Tarr, and Dundas displaying the binned aspect and slope distributions within each watershed.



**Tables**

### Table 1: Key physical characteristics of the 8 sub-watersheds in the study area.

| Watershed Name | Area (km²) | 2014 Glacier Coverage (%) | Mean Elevation (m) | Max Elevation (m) |
|---|---|---|---|---|
| GBNPP | 10085 | 37.7 | 584 | 4190 |
| North | 7824 | 33.9 | 657 | 3905 |
| West | 2261 | 51.0 | 790 | 4190 |
| West-Arm | 3098 | 54.2 | 1165 | 3905 |
| East-Arm | 2064 | 37.8 | 686 | 2216 |
| Tarr | 927 | 65.8 | 1453 | 3905 |
| Carroll | 793 | 68.1 | 897 | 2113 |
| Dundas | 386 | 17.6 | 331 | 1279 |

### Table 2: Summary of the SNAP selected climate models.

| Center | Model | Acronym |
|---|---|---|
| National Center for Atmospheric Research | Community Earth System Model 4 | NCAR-CCSM4 |
| NOAA Geophysical Fluid Dynamics Laboratory | Coupled Model 3.0 | GFDL-CM3 |
| NASA Goddard Institute for Space Studies | ModelE/Russell | GISS-E2-R |
| Institut Pierre-Simon Laplace | IPSL Coupled Model v5A | IPSL-CM5A-LR |
| Meteorological Research Institute | Coupled GCM v3.0 | MRI-CGCM3 |





**Table 3: Historic (1979-2015) and forecast (RCP 8.5 scenario; 2070-2099) runoff in km³ for all watersheds.**

| Watershed Name | Historic Runoff (km³) | Forecast Runoff (km³) |
|---|---|---|
| GBNPP | 34.2 | 40.0 |
| North | 24.5 | 27.5 |
| West | 9.7 | 12.4 |
| West-Arm | 10.6 | 13.4 |
| East-Arm | 7.5 | 7.5 |
| Tarr | 2.7 | 4.3 |
| Carroll | 2.3 | 2.3 |
| Dundas | 1.2 | 1.2 |





**Appendices**

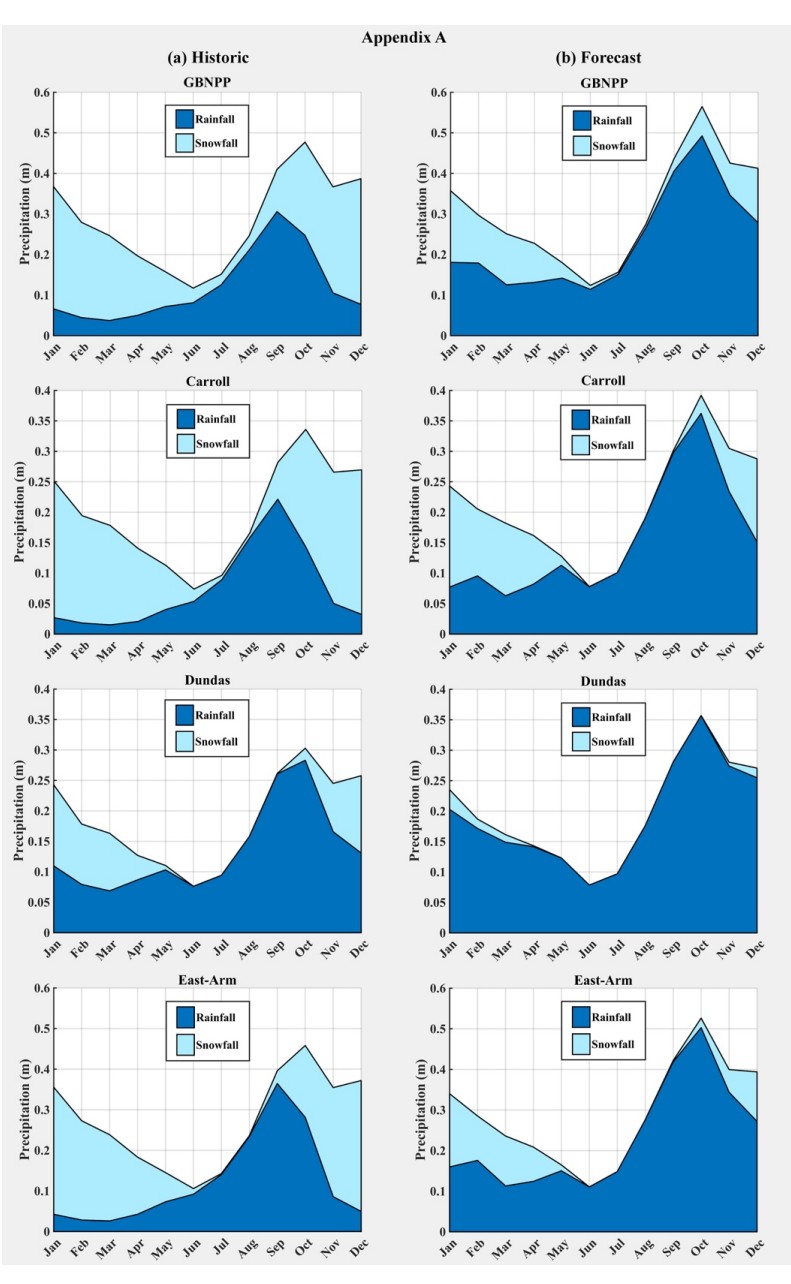





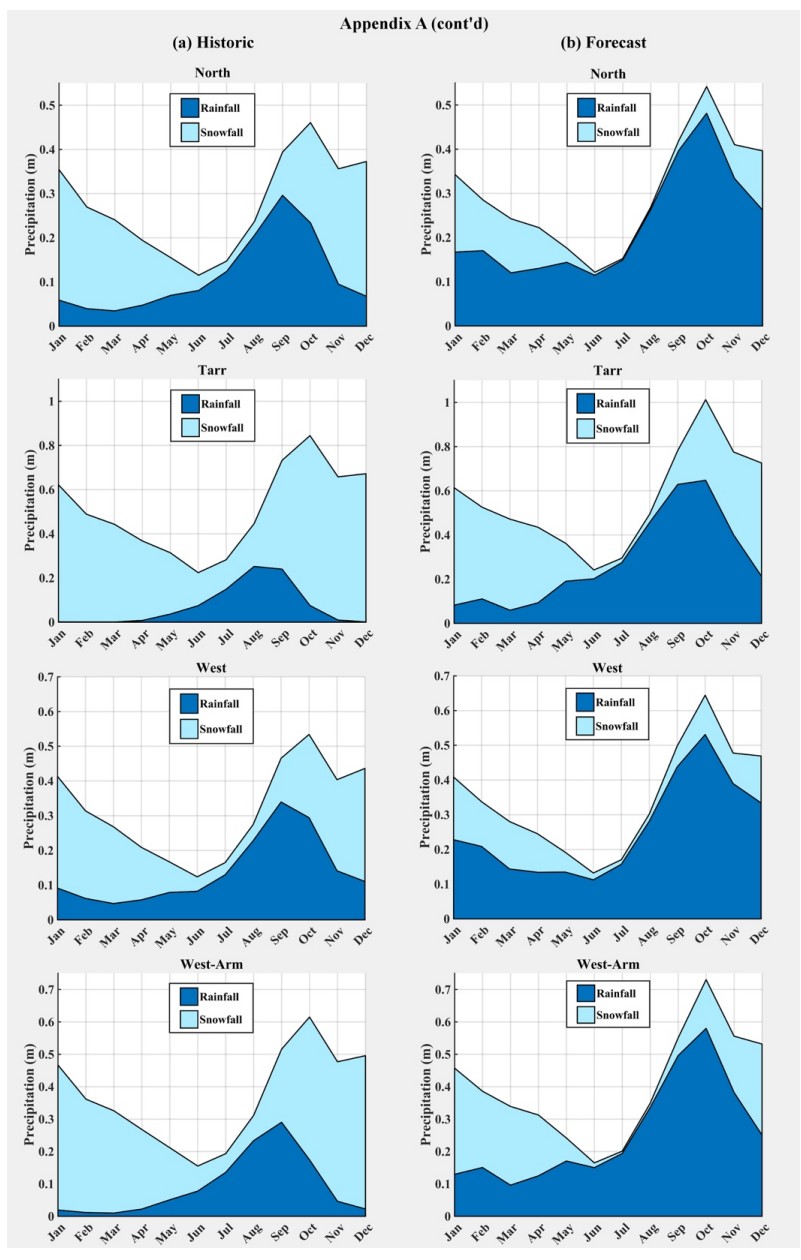

Appendix A. (a) – The historic precipitation climatologies by watershed, partitioned into snowfall and rainfall constituents. (b) – The forecast precipitation climatologies by watershed, partitioned into snowfall and rainfall constituents.





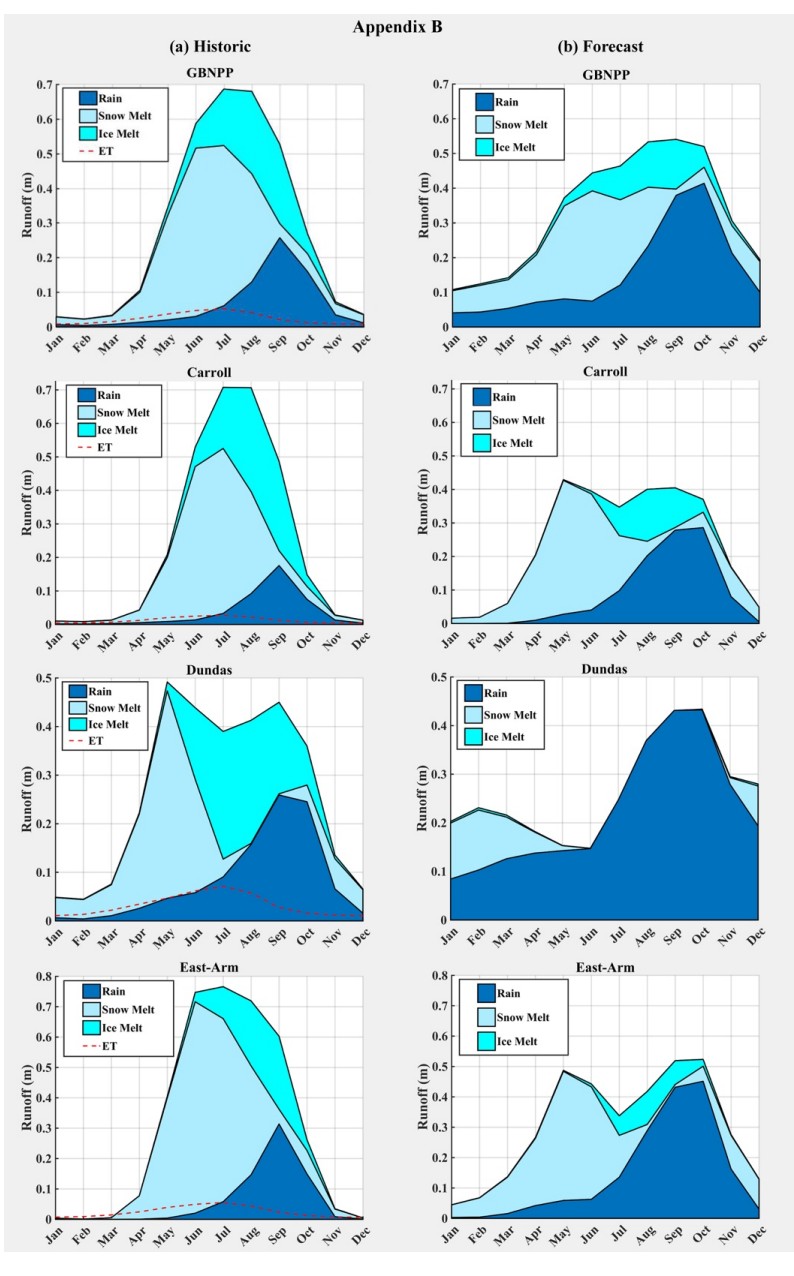





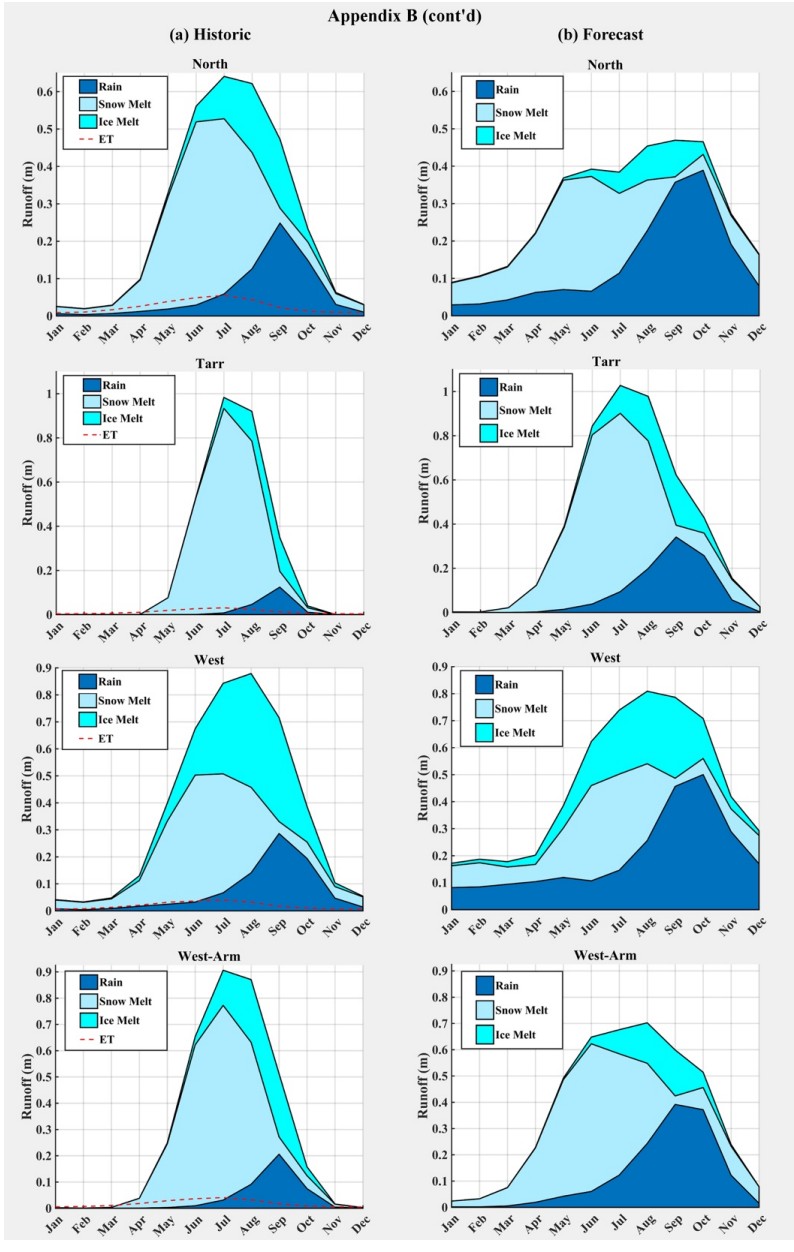

Appendix B. (a) – The historic runoff climatologies by watershed, partitioned into the constituents of snowmelt, ice- melt, and rain runoff. The historic MODIS-based evapotranspiration estimates are included on the historic plots, but the amounts are not subtracted from the modeled for runoff climatology because they were derived separately from the modeling process. (b) – The forecast runoff climatologies by watershed.