# Peer review of "Seasonal Components of Freshwater Runoff in Glacier Bay, Alaska: Diverse Spatial Patterns and Temporal Change"

_The Cryosphere, 2019_

## Referee Comment (RC1) · Kristian Förster (Referee) · 18 Feb 2019

**General comments**

In their manuscript the authors describe a very detailed study on the glacio-hydrological change in the Glacier Bay, Alaska. The paper is interesting, written very well and fits well into the scope of "The Cryosphere". The authors elaborated an energy balance snow model (SnowModel) which was extended by soil moisture and evapotranspiration in the framework of an earlier study. Elaborating energy balance approaches is timely and SnowModel is a suitable model to perform such type of analyses. A drawback is the static representation of glaciers. It is clear that dynamic glaciers are only represented in a small number of hydrological models. This topic is still subject to current research

(e.g., Seibert et al, 2018, Hanzer et al., 2018, or WaSiM which builds upon the work of Stahl et al, 2008). However, I would at least expect that this limitation should be discussed in more balanced way in the outlook section (e.g., utilizing more detailed approaches as future outlook).

Another limitation is the calibration of the model. In fact, the model applies "regional" parameters, which is sound, given that hydrological observations are not available in the study area. However, the discussion of the oceanographic data seems detached from the modelling experiment to a certain degree. Using this kind of data is really an asset in my opinion. Later, in Sect. 5, only a visual comparison is carried out. In my opinion, the comparison of the model with the oceanographic data (fresh water volume) could be considered in the calibration section (Sect. 3.4). Since this dataset is the only observational dataset available (and still very helpful!), you could consider showing its value in the calibration section. Even though this comparison is subjected to large uncertainties, addressing the model performance should be done quantitatively. The figure visualizing the trends in FWC could be moved to the Appendix (even though it is interesting, it is not clear to me why it is important in this context). Instead you could show a spatial representation of changes in SWE in order to better motivate the application of the very detailed energy balance snow melt model (250 m resolution which was mentioned to be an improvement compared to the existing work of Beamer et al., 2017). I did not fully understand the way how the scenarios are computed. I would encourage the authors to add some more explanations.

Overall, I think that the modelling experiment is sound and that the paper is a valuable contribution. Please find my comments below which might be helpful to improve the manuscript.

**Specific comments**

P1L17: Here, I would recommend to provide the annual runoff in mm per year too, since it helps to compare the values with other studies.

[Figure]

P1L23: Please provide an explanation for the abbreviation CTD. In the current version of the manuscript, it becomes only clear on page 5.

P3L9: What do you mean by "large uplift rates"? Please be more specific.

P3L16: Here, you explain that the model output is available as daily output. What is the internal time step of the (energy balance) model?

P4L25pp.: Here you could provide some more details on the time step of the model. Since it is an energy balance model, I would expect sub-daily time steps (even though the output is daily).

P5L15p.: ET is computed by SnowModel? Was there any attempt to compare the results with the MODIS data – at least for reasons of plausibility, given that there is a mismatch in scales between MODIS and SnowModel?

P5L27: A new subsection 3.3.1 is introduced in section 3.3 but there is not any other subsection (e.g., 3.3.2 etc.). I was wondering if it is worth to merge the sections 3.3 and 3.3.1?

P6L19: The term "forecast" is used throughout the manuscript to describe the scenario data and the corresponding results. I am not sure if this term is correct in this context. I would suggest using "projection" instead since this term acknowledges additional uncertainty involved in climate scenarios which arise from uncertain greenhouse gas emissions (i.e. external forcing that is not exactly known). For instance, in a recent paper we also used the term projection to highlight this type of forcing (Hanzer et al., 2018). In contrast, according to Kirtman et al. (2013), the term forecast refers to initialized climate model runs (e.g., seasonal to decadal predictions, see their Box 1.1, or http://glossary.ametsoc.org/wiki/Climate_prediction).

P6L30: Here, I would suggest to add some thoughts why you have selected RCP8.5 only. It is clear that running impact models for numerous RCPs is expensive in terms of computational costs. However, you could argue that you are interested in a worst-case

scenario to describe possible future changes.

P7L14: Does it mean that you did not apply SnowModel to future periods, e.g. by forcing the model with modified MERRA data (scaling of meteorological forcing)? From your explanations, you compared the historic run from SnowModel with the future Simulation of Beamer et al. (2017) in terms of long-term averages on runoff. If I understood this correctly, this would suggest a simple approach that contradicts the first line of your abstract ("… is used to estimate current and future runoff into Glacier Bay."). I would encourage the authors to provide more details on the setup of future scenarios.

P12L32p.: I was wondering why only temperature and precipitation have been considered, given that SnowModel requires additional meteorological quantities?

P12L36: The validation could be done in a quantitative way too. The only linkage between your results and oceanographic data is provided on page 12, lines 7 to 8 (by comparing Figure 7a with Figure 9). Since the model calibration is done for another region (indeed, in which your region is included), I would expect a closer look on this dataset, since it is the only dataset available for assessing model accuracy.

P22L9 (Figure 7): Why do you plot ET derived by MODIS only, given that your model accounts for ET too? If ET computations are available for the model too, you could plot ET for the future scenarios as well. In my opinion, analyzing changes in ET would be an interesting asset to describe the hydrological change.

P23L10 (Figure 9): Why do we see a maximum in delta FWV in January? I would at least expect a brief discussion on that maximum in the text.

P27 (Table 3): It would be helpful for the readers to have a separate column for each existing column which provides the runoff in mm too. In your text, you already highlight the benefit of using specific runoff for reasons of comparison.

**Technical comments**

P7L17: There is no Sect. 3.4.1.

P10L: Figure 9 does not show any trends. Why do you plot the delta in FWV instead of FWV?

P11L12: Please correct the reference to the figure (there is no Fig. 9a).

**References**

Beamer, J.P., Hill, D.F., McGrath, D., Arendt, A. and Kienholz, C.: Hydrologic impacts of changes in climate and glacier extent in the Gulf of Alaska watershed, Water Resources Research, 53, pp.7502-7520, doi:10.1002/2016WR020033, 2017.

Hanzer, F., Förster, K., Nemec, J. and Strasser, U.: Projected cryospheric and hydrological impacts of 21st century climate change in the Ötztal Alps (Austria) simulated using a physically based approach, Hydrology and Earth System Sciences, 22(2), 1593–1614, doi:10.5194/hess-22-1593-2018, 2018.

Kirtman, B., S.B. Power, J.A. Adedoyin, G.J. Boer, R. Bojariu, I. Camilloni, F.J. Doblas-Reyes, A.M. Fiore, M. Kimoto, G.A. Meehl, M. Prather, A. Sarr, C. Schär, R. Sutton, G.J. van Oldenborgh, G. Vecchi and H.J. Wang: Near-term Climate Change: Projections and Predictability. In: Climate Change 2013: The Physical Science Basis. Contribution of Working Group I to the Fifth Assessment Report of the Intergovernmental Panel on Climate Change [Stocker, T.F., D. Qin, G.-K. Plattner, M. Tignor, S.K. Allen, J. Boschung, A. Nauels, Y. Xia, V. Bex and P.M. Midgley (eds.)]. Cambridge University Press, Cambridge, United Kingdom and New York, NY, USA, 2013.

Seibert, J., Vis, M. J. P., Kohn, I., Weiler, M. and Stahl, K.: Technical note: Representing glacier geometry changes in a semi-distributed hydrological model, Hydrology and Earth System Sciences, 22(4), 2211–2224, doi:10.5194/hess-22-2211-2018, 2018. Stahl, K., Moore, R. D., Shea, J. M., Hutchinson, D. and Cannon, A. J.: Coupled modelling of glacier and streamflow response to future climate scenarios, Water Resources Research, 44(2), W02422 1-13, doi:10.1029/2007WR005956, 2008.

---

## Referee Comment (RC2) · Anonymous Referee #2 · 20 Feb 2019

Comments to the Author Summary of the manuscript This manuscript (ms) presents current and future fresh water runoff into Glacier Bay in Alaska in order to link it to current trends of the fresh water content in the fjords of Glacier Bay. This seems to be important as CTD measurements might indicate a change of fresh water content in the ocean water. For this purpose the authors used the high resolution "SnowModel" to calculate changes in runoff from three sub catchments in the Glacier Bay National Park and Preserve (GBNPP) group between a present time period and a future time period under climate change scenarios. The results indicate that fresh water inflow may decrease in spring but increase during the summer month under future climatic conditions. The study concludes that this is validated by the CTD observations in the bay.

[Figure]

Evaluation I think the topic of this manuscript is highly relevant and important in order to understand and anticipate climate change impacts on the sensitive marine ecosystem in Glacier Bay. Accordingly, I do think that this the topic of this ms should be considered for publication. However, I have my major doubts if the presented ms convincingly demonstrates that the climate projections based on hydrological modeling can explain the CTD observations. My main concerns are the following: i) the link between modelling results and CTD observations is weak, lacks description and a convincing discussion. ii) the calibration of the modelling chain lacks description and convincing results. iii) an uncertainty discussion of the modelling results is missing. iv) the structure of the ms is at some locations mixing methods, data description and results v) due to the concerns above the conclusions are vague, speculative and lack conviction.

I leave it up to the editors to decide if the present ms can be revised or should be resubmitted. I would recommend to address the following concerns prior to publication:

Major concerns: 1. The linkage between FWV and FWC would require a thorough discussion of the mixing of freshwater into ocean water. Numerous paper exist on this topic but this ms fails to review the literature and discuss this complex topic in a convincing manner. 2. The description of the calibration is weak: if the authors claim to make realistic projections of future FWV into the ocean, I would expect a thorough discussion of the efficiency regarding snow melt, ice melt and rain runoff of the model; the necessity of multi dataset calibrations for hydrologic modelling under climate scenarios have been discussed in the literature. 3. The uncertainty of the results are not discussed; are the projected changes significant? What is the uncertainty of the future scenarios?

Specific comments: 1) Title: what is hydrologic diversity? This term is never mentioned in the ms accordingly it seems misleading to use it in the title. I would be helpful to have a title that reflects the content of the ms. 2) Abstract: An introductory sentence explaining the problematic and the purpose of the ms is missing; L16 why "wide variety", the same "variety" exist in any glaciated catchment; L24: this sentence is redundant, as

it does not contain any conclusive information about the study: 3) Introduction: Nice description of the study site; however, a description of the linkage between fresh water inflow in an ocean bay and the subsequent impacts on marine life is missing; also a review of the literature of intruding freshwater into water bodies would be helpful. pg3, L 15: "the goals are different": it would be helpful to outline the goals; L20: the results present do not convincingly present changes in the coastal runoff (see major concerns). 4) Methods1: pf4, L 8: model chain? Only two models are used, one for the reanalysis of the forcing data and the SnowModel. 5) Methods2: I think the clarity of the ms would improve if methods and data were two separate chapter; 6) Methods 3: 3.4. describes in a very rudimentary way model calibration; r2 and NSE values are provided. Since the authors claim to provide an "added understanding" and "constrained estimates of how costal runoff will change in the future" I would expect a thorough discussion on the efficiency of their modelling in regard to runoff, snow melt and ice melt contribution during the calibration period. If the calibration is not presented adequately, how can one trust in the results of future runoff? 7) Results: pg7,8: here results and methods to calculate the results are in the same chapter; I think a clear separation between methods and results would be helpful; 8) Figures 1,2 and 5 (in total 8 maps) all show specific aspects of the study site; this information could be combined and presented in one or perhaps 2 large panels. 9) Figure 3: I do not understand why contour plots are used here; bars indicating the exact value of T and P change would be more helpful. 10) Figures 4, 6, 7, 8, 9 and 10: it would be helpful to add an uncertainty to each point; e.g. stdev from the mean over the 30 yrs (but this would only account for climatic availability); I recommend checking recent literature on this topic.

---

## Referee Comment (RC3) · Janet Curran (Referee) · 28 Feb 2019

Review of Crumley et al. Hydrologic Diversity in Glacier Bay

Summary and general comments

This manuscript presents the results of high spatial-resolution hydrologic modeling of a dynamically deglacierizing environment, Glacier Bay in Alaska. The study furthers understanding of ecological parameters in Glacier Bay and freshwater runoff to the Gulf of Alaska by estimating changes in seasonal distribution of runoff between a historical and a future scenario and attributing changes to shifts between the runoff-producing processes of rainfall, snowmelt, and ice-melt. This topic is timely, relevant to current research questions in this field and in this geographic area, and suitable for this journal.

The text is well written, the figures and tables are all appropriate and useful, and the details included in the Appendices are appreciated. This study relies heavily on methods developed for companion studies by several of the authors (especially Beamer et al., 2016 and 2017). Those robust studies are well-supported by references and descriptions in the text. Unfortunately, the present study suffers from the complete lack of available calibration data, which the authors overcome through (1) adoption of the model choices of the companion studies and (2) general comparisons to oceanographic salinity data. This is a creative solution and acceptable for the goals of the study but should be presented much earlier and more plainly in Section 3.4 Model Calibration. After reading author contributions, the disconnects in the presentation and significance of the oceanographic data seem more a matter of author coordination than flaws in the study and can be resolved without re-analysis.

Specific comments

Title: Can you find a term other than "hydrologic diversity" that better brings the topics of changes in runoff volume, seasonality, and drivers to mind? Hydrologic regime diversity...? Freshwater runoff ...? Not sure I have the perfect term, might take a phrase to say it.

Abstract, L 24-25: "a variety of changes" is vague. What is meant here?

P6, L15-17: The closest calibration point isn't always the most appropriate. Can you also say that the Mendenhall basin is the most similar?

P6, L17-18: These metrics are for the Beamer et al. (2016) study, correct? Since this "calibration" section oddly refers to the calibration of a prior study, I suggest phrasing this clearly so the skimming reader doesn't assume these metrics are for your study.

P8, L1: This sounds like a justification for a higher-than-expected result, but the wording isn't clear. Does the RCP8.5 scenario establish a minimum of 3 degrees change?

P10, L3: Is the 3.40 m/yr value actually for "runoff", not "precipitation" as stated? That

would be more consistent with the value in the next line.

P12, 13-14: This statement about a non-stationary system is inconsistent with the presentation of Figure 10 on P10, L20-22, which notes little significant change with one basin as an exception. The trend for the excepted basin isn't very convincing (p > 0.05, short and discontinuous dataset, a bit noisy), making the comments on P12 seem overstated.

Appendix B: Interestingly, the forecast runoff hydrographs, which admirably show the relative contributions of runoff processes, produce a few seasonalities that aren't apparent for individual streams in my present work characterizing historical hydrographs. The composite GBNPP and North basins appear to have a snowmelt-dominated spring peak and a larger rainfall-dominated fall peak, a reversal of the typical relative magnitudes for a bimodal glacierized basin hydrograph. Can this be explained by an increase in spatial distribution of future rainfall-dominated areas within the composite basin or any other observations from the modeling?

Technical comments

Introduction: Trim and keep focused on the study by omitting details about GOA (especially in 1st and 2nd paragraph), minimizing drama (P2, L6-7), and considering moving setting information to the Study Area section if it's actually needed (the long discussion of tidal mixing and stratification and the Etherington et al. study made me think this was the study focus on first read). It's all interesting, but it's not until the penultimate paragraph (P3, L37-39) that the problem is hinted at and not until the final paragraph (P3, L13) that the actual work of the study is introduced, and the reader can finally start understanding the direction of the manuscript.

P2, L31 and P3, L3-4: The number of references to particular places within Glacier Bay suggests Figure 2 could be presented earlier.

Study Area, paragraphs 1 and 2: Clearly define study area (all watersheds within GB-

NPP, which includes all the lands of GBNPP and some areas outside it?). The multiple nested, paired watersheds are a nice study design but are hard to keep track of. Suggest moving the parts of paragraph 2 that aren't obvious from the figure or table (P3, L29-30) into paragraph 1. Consider using a defining characteristics for the names or adding a column to Table 1 to associate basin names with a defining characteristic. It would be helpful to know "North" is the full Glacier Bay basin and that the choice of the three named basins allows comparison of basins having. . .(a range of elevation? a range of glacier characteristics?), for example.

P7, L23-24: This is one of the clearest statements of the goals/outcome of the study. Could use this earlier.

P9, L11-12: Delete information repeated from methods.

P9, L25-35: Many details of computations, and the discussion of the omission of routing, seem like methods. Consider moving to Section 3.1 or elsewhere in Methods.

P10 and 11, Section 5, first and second paragraphs: Most of the main points are made in the first paragraph; suggest combining the two and reducing detail. Consider moving computation of FLAs to methods.

P12, L5-6: Nice explanation of why CTD dataset was included, could use this earlier.

P11, L11-12 and L20: Check figure number. I assume you mean figure 11a and b, respectively.

References: References are used appropriately. I did not check to make sure all are used, or that all references cited are included. The recommended citation for USGS reports includes the report series title and report number. For Curran et al. (2003) that's Water-Resources Investigations Report (or WRIR, if preferred) 03-4188 and for Wiley and Curran (2003) it's Water-Resources Investigations Report 03-4114.

Fig. 1: Labeling Alaska and Canada (a) and Glacier Bay (b and c) would help reader comprehension.

Fig. 2: Label Glacier Bay. The Alaska/Canada boundary is referenced in the text but not shown here.

Fig. 5 : Shading of forecast glaciers is distractingly similar to ocean. The title "Glacier Change" doesn't match the legend items, which include two glacier positions and the GBNPP boundary.

Fig. 6: Suggest being consistent with the x-axis scale used for other monthly plots (use Jan-Dec, not Sept-Aug)

Fig. 7 caption, last sentence for (a): Check for typo in "the modeled for runoff climatology"

Tables 1 and 3, and Appendix A and B: Suggest some structure to convey basin/sub-basin relationship and the various pairings of nested basins (a line or spacing, for example). At a minimum, keep the same order in the Appendices as is used for the tables.

Appendices: These plots are useful results and would lend themselves well to being reduced in size. Consider rearranging to fit each Appendix on 1 page with a single legend for each and including in the text.
* * *

---

## Author Comment (AC1) · 7 May 2019

Reviewer 1: Kristian Förster

**P1L17: Here, I would recommend to provide the annual runoff in mm per year too, since it helps to compare the values with other studies.**

Response: This is a good suggestion, the values in m/yr have been added to the abstract and the conclusions in the final draft.

**P1L23: Please provide an explanation for the abbreviation CTD. In the current version of the manuscript, it becomes only clear on page 5.**

Response: A correction for this abbreviation has been added to the final draft.

**P3L9: What do you mean by "large uplift rates"? Please be more specific.**

Response: We are referring to the regional uplift rates for Glacier Bay and southeast Alaska from isostatic rebound caused by glacial wastage since the little ice age. See Larsen et al. (2005) for more information. For clarity, in Section 1, ¶6 we changed large to *rapid* and added *from isostatic rebound* to the final draft.

Larsen, C.F., Motyka, R.J., Freymueller, J.T., Echelmeyer, K.A. and Ivins, E.R., 2005. Rapid viscoelastic uplift in southeast Alaska caused by post-Little Ice Age glacial retreat. *Earth and Planetary Science Letters*, *237*(3-4), pp.548-560.

**P3L16: Here, you explain that the model output is available as daily output. What is the internal time step of the (energy balance) model? P4L25pp.: Here you could provide some more details on the time step of the model. Since it is an energy balance model, I would expect sub-daily time steps (even though the output is daily).**

Response: The internal timestep of the model is 3 hrly and the results have been aggregated to daily and monthly for the climatological analysis. A sentence about the timestep information has been added to the model Section 3.1, ¶3.

**P5L15p.: ET is computed by SnowModel? Was there any attempt to compare the results with the MODIS data – at least for reasons of plausibility, given that there is a mismatch in scales between MODIS and SnowModel?**

Response: ET of the land surface is not calculated by SnowModel. When snow is present in the grid cell, sublimation of the snowpack is calculated by the energy balance sub-model (EnBal) and sublimation of blowing snow is calculated by the snow transport sub-model (SnowTran-3D), see Liston et al. (2006) for a review of all of the model physics and subroutines. It should be noted

that the snow transport model is not recommended for model resolutions above 100m, and since our model resolution is 250m, SnowTran3-D was not utilized for the simulations.

However, obviously ET does make up a portion of the water balance and we therefore use the MODIS ET dataset to supplement the results of our SnowModel simulations. Hill et al. (2015) estimated the ET component of annual runoff for the entire Gulf of Alaska region to be ~17%. Beamer et al. (2016) estimate ET for the Gulf of Alaska region to be 10-15% less than Hill et al. (2015) results. Since much of the GBNPP domain is glaciated or covered in snow for many months of the year, the authors decided to simply estimate ET values from the MODIS ET dataset for the historic simulation time period and spatially subset by watershed or grouped watershed. See section 3.2.3 for more information, as well as the section below in our responses regarding another ET question by Reviewer1. We find that the MODIS based ET values range from 5%-13% of annual runoff in the GBNPP watersheds and we've added a table with this information below. The authors decided to add this table (new Table 4) to the manuscript to clarify the ET process and results.

| Watershed Name | Historic MODIS ET (m/yr) | Percentage of Annual Precipitation (%) | Adjusted Annual Runoff (m/yr) |
|---|---|---|---|
| GBNPP | 0.3 | 9 | 3.1 |
| North | 0.3 | 9 | 2.8 |
| West | 0.2 | 5 | 4.1 |
| West-Arm | 0.2 | 5 | 3.2 |
| East-Arm | 0.3 | 9 | 3.3 |
| Tarr | 0.2 | 3 | 2.7 |
| Carroll | 0.2 | 8 | 2.7 |
| Dundas | 0.4 | 9 | 2.7 |

Beamer, J.P., Hill, D.F., Arendt, A. and Liston, G.E., 2016. High-resolution modeling of coastal freshwater discharge and glacier mass balance in the Gulf of Alaska watershed. *Water Resources Research*, *52*(5), pp.3888-3909.

Hill, D.F., Bruhis, N., Calos, S.E., Arendt, A. and Beamer, J., 2015. Spatial and temporal variability of freshwater discharge into the Gulf of Alaska. *Journal of Geophysical Research: Oceans*, *120*(2), pp.634-646.

**P5L27: A new subsection 3.3.1 is introduced in section 3.3 but there is not any other subsection (e.g., 3.3.2 etc.). I was wondering if it is worth to merge the sections 3.3 and 3.3.1?**

Response: This is a good idea, and we merged these two sections in the final draft.

**P6L19: The term "forecast" is used throughout the manuscript to describe the scenario data and the corresponding results. I am not sure if this term is correct in this context. I would suggest using "projection" instead since this term acknowledges additional uncertainty involved in climate scenarios which arise from uncertain greenhouse gas emissions (i.e. external forcing that is not exactly known). For instance, in a recent paper we also used the term projection to highlight this type of forcing (Hanzer et al., 2018). In contrast, according to Kirtman et al. (2013), the term forecast refers to initialized climate model runs (e.g., seasonal to decadal predictions, see their Box 1.1, or http://glossary.ametsoc.org/wiki/Climate_prediction).**

Response: The difference in these terms is important, and the authors agree that the term 'projection', as defined by Reviewer1 above, is more in line with the intention of the use of 'forecast' in the original manuscript. We are not attempting to make a climate prediction (as defined in the provided weblink) of what will happen in the future in Glacier Bay. We are modeling one of the potential scenarios that may occur in the hydrology of the region that would accompany the RCP8.5 emissions scenario. For these reasons, the authors have chosen to change the term 'forecast scenario' to 'projection scenario', in every instance, throughout the manuscript. These changes are added to the final draft.

**P6L30: Here, I would suggest to add some thoughts why you have selected RCP8.5 only. It is clear that running impact models for numerous RCPs is expensive in terms of computational costs. However, you could argue that you are interested in a worst-case.**

Response: See Section 3.5.1, ¶2 for text added as an explanation for why we chose the RCP8.5 scenario. The modified paragraph is below, added text is italic.

"Although future climate simulations *from SNAP* exist for numerous RCP (representative concentration pathway) scenarios, in this study we restrict ourselves to the RCP 8.5 scenario and to the 5-model mean. *The other RCP scenarios (RCP 2.5, RCP 4.5, RCP 6.0) represent concentrations of greenhouse gases (GHGs) in the atmosphere that peak earlier in the 21st Century or at lower levels of GHGs than the RCP 8.5 scenario. Keep in mind that the choice of the RCP 8.5 scenario is not an attempt to evaluate the likelihood of the future GHG concentrations. Rather, we use the RCP 8.5 scenario for the projection scenario because we are interested in the hydrologic changes that might occur in the worst-case scenario.*"

**P7L14: Does it mean that you did not apply SnowModel to future periods, e.g. by forcing the model with modified MERRA data (scaling of meteorological forcing)? From your explanations, you compared the historic run from SnowModel with the future simulation of Beamer et al. (2017) in terms of long-term averages on runoff. If I understood this correctly, this would suggest a simple approach that contradicts the first line of your abstract (". . . is**

**used to estimate current and future runoff into Glacier Bay.").** **I would encourage the authors to provide more details on the setup of future scenarios.**

Response: In section 3.5.3 Future Climatologies, we discuss creating the projection scenario climatologies. First, a review of the process. Beamer et al. (2017) conducted a SnowModel historic and forecast simulation for the larger Gulf of Alaska study area. The authors subset the GBNPP study area results from the more spatially extensive Gulf of Alaska simulations. These SnowModel simulations were forced with CFSR for the historic and projection scenarios, for the entire model space (at the lower 1km resolution) and complete model timeframe (3hrly). Climatologies were created by temporal averaging and spatial aggregation into the GBNPP watersheds or grouped watersheds.

Next, a full SnowModel historic simulation for the GBNPP study area was conducted at the higher spatial resolution of 250m, forced by the MERRA reanalysis product. Climatologies were then created by temporal averaging and spatial aggregation by watershed or grouped watershed. At this point, we have spatially and temporally averaged climatologies for each watershed or grouped watershed (1. CFSR-based historical climatologies, 2. CFSR-based projection scenario climatologies, 3. MERRA-based historical climatologies). We created a scaling factor from the CFSR historic and projection scenarios to apply to the MERRA historic climatologies. After the application of these scaling factors we have the MERRA-based projection scenario climatologies by watershed or grouped watershed.

This study presents projection scenario results as 30-year climatologies. We do not present results related to the frequency characteristics of the runoff. As such, we are not presenting results on frequency distributions, or peak flows, etc. For long-term characteristics of runoff, however, we believe that our approach is appropriate because changes in runoff are driven by long-term changes in precipitation and temperature, which vary relatively slowly in space, and these changes are preserved in the scaling from CFSR-based historical climatologies to CFSR-based projection scenario climatologies.

**P12L32p.: I was wondering why only temperature and precipitation have been considered, given that SnowModel requires additional meteorological quantities?**

Response: This is a decision based in part on the fact that only temperature and precipitation anomalies from SNAP for the RCP scenarios are available for the AK region. While MicroMet inputs include relative humidity, wind speed, wind direction, shortwave and longwave radiation, and surface pressure, those have not been modified in the projected scenario. In order to investigate the impacts of leaving out any potential changes in other variables, such as humidity, the authors considered the changes in relative humidity for their Gulf of Alaska results and the results from this additional analysis are below.

First, it's important to understand that MicroMet will modify the radiation balance through temperature increases. See the figure below, where its obvious that longwave radiation changes substantially with both Temperature and Relative Humidity. Since we already have a good idea of the expected increases in Temperature from SNAP (from the ΔTemp analysis discussed in our Results Section 4), the question becomes how different might relative humidity values be in the projection scenario?

[Figure]

To answer this question, we turned to the VEMAP project (https://daac.ornl.gov/VEMAP/guides/VEMAP_Alaska.html) which provides low resolution downscaled monthly grids of many variables including relative humidity. These grids come from two climate models, and we looked at the results from one (HadCM2). The results provided were for the GHG+A1 scenario, sometimes referred to IS92a. This scenario is quite comparable to the SRESA2 scenario which, in turn, falls between the RCP6.0 and RCP8.5 projection scenarios. We computed climatologies of relative humidity for the periods 1966-1996 and 2070-2100 for four months of the year (Jan, Apr, Jul, Oct). The figure below shows the differences in relative humidity between the two climatological periods (Future – Historical).

[Figure]

In the figure above, we see that Relative Humidity is projected to increase throughout the Alaska region, including the Glacier Bay region in the lower, right portion of the figure. In July and October, the increases are typically less than five percent. In January, the increases throughout Alaska are in the range of 5-10 percent. The first figure above, which shows longwave radiation as a function of temperature and RH, suggests that the ~4 deg C temperature increases that are predicted by RCP 8.5 produce a change in longwave of about 25 watts per square meter. An increase in RH of about 5-10 percent appears to produce a smaller (~10 watts per square meter) change in longwave. So, while it would be conceptually preferable to adjust all weather forcing variables, it appears that the expected changes in temperature dominate. This fact is what led us to adopt the future SNAP climatologies for Temperature and Precipitation as a good 'leading order' study of the effects of changing climate with the RCP8.5 projection scenario.

Additionally, the focus of this manuscript and scientific questions is not changes in climate variability, weather extremes, or high runoff events, but rather longer-term, climatological averages. The authors think it's not within the scope of this project to create our own statistically downscaled, projection scenario weather variables (RH, shortwave and longwave radiation). Since we are primarily focused on the climatological averages of the model output, we find the choice of perturbing the temperature and precipitation inputs for the projection scenario to be adequate, and aligned with the methods summarized in Beamer et al. (2017).

**P12L36: The validation could be done in a quantitative way too. The only linkage between your results and oceanographic data is provided on page 12, lines 7 to 8 (by comparing Figure 7a with Figure 9). Since the model calibration is done for another region (indeed, in which your region is included), I would expect a closer look on this dataset, since it is the only dataset available for assessing model accuracy.**

Response: The authors decided to use the oceanographic dataset primarily in a qualitative way because the of the complex, understudied open boundary of the bay system, where water (fresh and salt) moves freely in and out of the boundary into Icy Strait, the Cross Sound, and eventually the Pacific Ocean. Critically, freshwater fluxes are not measured or analyzed at the mouth of Glacier Bay, and to the best of the authors' current knowledge, a dataset that includes these fluxes entering and exiting the system does not exist. Thus, we are not able to explicitly determine what component of FWV is sourced by freshwater runoff as opposed to fluxes of highly stratified water at the bay's inlet. Therefore, we've chosen to focus on the change in freshwater volumes from month to month within the bay, instead of quantifying total bay freshwater volume at any given time. Given the sharp temporal gradients in freshwater runoff, the oceanographic dataset is presented to qualitatively assess whether the FWV signal shows a strong temporal gradient that is synchronous with freshwater runoff predicted by the model.

**P22L9 (Figure 7): Why do you plot ET derived by MODIS only, given that your model accounts for ET too? If ET computations are available for the model too, you could plot ET for the future scenarios as well. In my opinion, analyzing changes in ET would be an interesting asset to describe the hydrological change.**

Response: In Figure 7 and Appendix A we plot monthly ET values derived from MODIS because SnowModel calculates sublimation of the snowpack when solving the energy balance equations but does not calculate ET from the land surface when no snowpack is present. See previous answer on ET for more details. This is why Beamer et al. (2017) added the SoilBal sub-model to their analysis of the Gulf of Alaska SnowModel simulations. Many other previous studies using SnowModel from the Arctic, Patagonia, Greenland, and Alaska do not calculate ET using an additional sub-model, and this manuscript is no different. (Mernild et al., 2012; 2013; 2014; 2017a)

Our intention in plotting the MODIS derived ET values on the runoff plots (Figure 7 and Appendix A) is to give the reader an estimation of how much the monthly runoff would be altered if ET were included. See the explanation in section 3.2.3 for more details. We temporally average (14-year), and spatially aggregate the MODIS ET values for each watershed/grouped watershed. We do not subtract these monthly values from the partitioned climatologies (snowmelt, glacier melt, rain runoff) because the model does not resolve which of these sources would be the appropriate origin of the ET-based water. Even more importantly, in the projection scenario, we have no land cover evolution beyond a simple estimation of glacier area change. If glaciers continue to recede, as they have over the last several hundred years and are projected to change in the future in Glacier Bay, these changes will continue to alter the landscape, in both landcover species and landcover type

designations. These landcover changes would inevitably cause changes to ET in the future. We admittedly make no attempt to quantify or characterize these types of landcover changes, nor the subsequent ET changes, because it is outside the scope of our current project and analysis.

To clarify the ET results, the authors will be adding this table in the final draft of the estimated monthly ET values from MODIS by watershed and we include the percentage of annual runoff and the adjusted annual runoff values. We have also added the adjusted runoff values in the abstract and conclusion paragraphs for clarity. Note: these are historic values because we do not estimate ET for the projection scenario.

| Watershed Name | Historic MODIS ET (m/yr) | Percentage of Annual Precipitation (%) | Adjusted Annual Runoff (m/yr) |
|---|---|---|---|
| GBNPP | 0.3 | 9 | 3.1 |
| North | 0.3 | 9 | 2.8 |
| West | 0.2 | 5 | 4.1 |
| West-Arm | 0.2 | 5 | 3.2 |
| East-Arm | 0.3 | 9 | 3.3 |
| Tarr | 0.2 | 3 | 2.7 |
| Carroll | 0.2 | 8 | 2.7 |
| Dundas | 0.4 | 9 | 2.7 |

Mernild, S.H. and Liston, G.E.: Greenland freshwater runoff. Part II: Distribution and trends, 1960-2010, *Journal of Climate*, *25*(17), pp.6015-6035, DOI: 10.1175/JCLI-D-11-00592.1, 2012.

Mernild, S.H., Lipscomb, W.H., Bahr, D.B., Radić, V., Zemp, M.: Global glacier changes: a revised assessment of committed mass losses and sampling uncertainties, The Cryosphere 7, 1565–1577, DOI: 10.5194/tc-7-1565-2013, 2013.

Mernild, S.H., Liston, G.E. and Hiemstra, C.A.: Northern hemisphere glacier and ice cap surface mass balance and contribution to sea level rise, *Journal of Climate*, *27*(15), pp.6051-6073, DOI: 10.1175/JCLI-D-13-00669.1, 2014.

Mernild, S.H., Liston, G.E., Hiemstra, C.A., Malmros, J.K., Yde, J.C. and McPhee, J.: The Andes Cordillera. Part I: snow distribution, properties, and trends (1979–2014), *International Journal of Climatology*, *37*(4), pp.1680-1698, DOI: 10.1002/joc.4804, 2017a.

**P23L10 (Figure 9): Why do we see a maximum in delta FWV in January? I would at least expect a brief discussion on that maximum in the text.**

Response: This is a good question, and we decided to look at these specific months (Jan and Dec) in our analysis below. For clarity, the January value ΔFWV_jan in Figure 9 is equal to FWV_jan – FWV_dec. In assessing the certainty in this signal, it is important to consider: (1) Winter is vastly under-sampled as compared to other seasons, and (2) there can be great variability between monthly FWC from year to year. The authors realized that because of these two factors, some months were excluded in the FWV extrapolation for all months. After incorporating these excluded months into the analysis, we made changes to Figure 9 that is more representative of all months in the entire dataset. See the old and new versions of Figure 9 below. Some of undersampled winter monthly FWV values have changed slightly, the more highly-sampled summer monthly FWV values remain largely intact, and the ΔFWV_jan value is dampened in the final Figure 9. In the revised figure, it is evident that the strongest signals in monthly changes in freshwater occur in the summer (May to October). Although the figure does suggest that there is an increase in FWV from December to January, it must be weighed in the context of the uncertainties described herein.

[Figure]

**Old Figure 9**

[Figure]

**New Figure 9**

**P27 (Table 3): It would be helpful for the readers to have a separate column for each existing column which provides the runoff in mm too. In your text, you already highlight the benefit of using specific runoff for reasons of comparison.**

Response: A new column has been added to Table 3 in the final draft with the specific runoff in meters for the historic and projection scenarios.

**P7L17: There is no Sect. 3.4.1.**

Response: Section 3.4.1 has been removed in the final draft.

**P10L: Figure 9 does not show any trends. Why do you plot the delta in FWV instead of FWV?**

Response: The word *trends* in Section 4.6, ¶1 has been changed to *seasonal timing of changes in freshwater*, which is a more precise wording of the sentence.

**P11L12: Please correct the reference to the figure (there is no Fig. 9a).**

Response: This mistake has been removed in the final draft.

---

## Author Comment (AC2) · 7 May 2019

Reviewer 2: Anonymous

**My main concerns are the following: i) the link between modelling results and CTD observations is weak, lacks description and a convincing discussion. ii) the calibration of the modelling chain lacks description and convincing results. iii) an uncertainty discussion of the modelling results is missing. iv) the structure of the ms is at some locations mixing methods, data description and results v) due to the concerns above the conclusions are vague, speculative and lack conviction.**

**Major concerns: 1. The linkage between FWV and FWC would require a thorough discussion of the mixing of freshwater into ocean water. Numerous paper exist on this topic but this ms fails to review the literature and discuss this complex topic in a convincing manner.**

Response: The manuscript originally contained a more detailed explanation of oceanographic processes, with a more thorough discussion of freshwater mixing. However, it is not within the scope of this manuscript to review in great detail the theoretical extent of these oceanographic processes, nor is it especially relevant to the readers of The Cryosphere. Instead, we added a paragraph in Section 1 in the manuscript that focuses on the bioecological effects of stratification in Glacier Bay specifically, in order to address this comment and to keep the focus on the study area. The paragraph added to the intro section in the final draft of the manuscript is below in italics.

*Long-term shifts in terrestrial freshwater storage and runoff can have significant implications for oceanographic stratification and circulation that moderate biogeochemical and ecological activity within Glacier Bay. Since Glacier Bay is a highly understudied, relatively remote national park, the complete freshwater budget for the bay cannot be quantified due to the lack of available data. However, seasonal trends in modeled freshwater runoff can be qualitatively compared with seasonal trends in broadscale oceanographic salinity records from 1993 to present collected by the U.S. National Park Service's Southeast Alaska Inventory & Monitoring Network (SEAN). This SEAN dataset served as the basis of the analysis performed by Etherington et al. (2007), which found positive correlations between phytoplankton biomass and stratification levels. The competing forces of macro-tidal flushing and strong stratification within the glacially-carved estuary generates temporally and spatially shifting trends in upwelling and nutrient availability (Etherington et al., 2007). Thus, accurate estimation of projection scenario runoff into Glacier Bay plays a paramount role in constraining future changes in water and nutrient circulation.*

**2. The description of the calibration is weak: if the authors claim to make realistic projections of future FWV into the ocean, I would expect a thorough discussion of the efficiency regarding snow melt, ice melt and rain runoff of the model; the necessity of multi dataset calibrations for hydrologic modelling under climate scenarios have been discussed in the literature.**

Response: This comment from Reviewer 2 about the calibration description is warranted, and in the light of other reviewer's comments, the authors added a paragraph in Section 3.4 to clarify our calibration decisions and process. It should be noted that one of the primary reasons the authors chose to use the historical oceanographic dataset is because long-term observational datasets of stream flow and weather conditions within the boundary of Glacier Bay National Park do not exist. This is also why we chose to calibrate the model with observations and records from the nearby Mendenhall Glacier, as further explained in more detail in the new paragraph in Section 3.4. The paragraph added to the calibration section in the final draft of the manuscript is below in italics.

*Recent studies (Beamer et al., 2016; Lader et al., 2016) have investigated the accuracy and biases of the MERRA reanalysis product in coastal Alaska compared to other reanalysis products such as ERA-Interim (Dee et al., 2011), CFSR (Saha et al., 2010), NCEP-NCAR (Kalnay et al, 1996), NARR (Mesinger et al., 2006), and others. Many SnowModel parameters were tested by doing a sensitivity analysis for each reanalysis product, including monthly precipitation adjustment factors, snow/rain temperature thresholds, snow and ice albedo factors, and more (see Beamer et al. (2016) their Table 2). For each of 4 reanalysis products, they calibrated model parameters based on observations of streamflow (Q) and glacier mass balance (B). The MERRA simulation Coefficient of Determination scores (r2) for glacier mass balance (B) and stream discharge (Q) for the Beamer et al. (2016) study were 0.80 and 0.95, respectively, and the Nash Sutcliffe Efficiency (NSE) scores were 0.67 and 0.91, respectively. While Beamer et al. (2016) identified the CFSR product as the 'best overall' for the GOA region, they found that MERRA was superior at the Mendenhall Glacier observational station, which is the closest calibration point (< 25 km) to GBNPP. For these reasons, in this study we rely on the model calibration of Beamer et al. (2016; their section 3.4) and we adopt their calibration parameters for SnowModel from their Table 2 and Table A1.*

*Long-term glacial mass balance programs and long-term streamflow gauge datasets do not exist within the GBNPP study area, thus constraining our ability to conduct additional calibration efforts. While the Mendenhall Glacier observation station is close in proximity to Glacier Bay, the glacier has receded and thinned significantly since the early 1900's, glacial wastage is a significant component of annual streamflow (17%), and glacial meltwater contributes heavily to streamflow in the summer (50%; Motyka et al., 2003). As a result of these similarities in geography and hydrology, we rely on the calibration process, parameters, and best-performing reanalysis product (MERRA) from Beamer et al. (2016) for our study.*

**3. The uncertainty of the results are not discussed; are the projected changes significant? What is the uncertainty of the future scenarios?**

Response: Please see the response to another comment from this Reviewer below about uncertainty and the variability of the historical simulations. Here, the Reviewer raises an important point, and included below is an analysis of the variability of the historical simulation results for the all watersheds in GBNPP. This table includes historical annual variability (+/- $\sigma$ standard deviation

in the table below) for the GBNPP study area and various grouped watersheds for the modeled output for runoff, snow precipitation, and SWE. This table includes the monthly climatological average (36-year) of the variable, the standard deviation of the 36-year monthly values of the variable, and the percentage of the climatological average represented by the +/- one standard deviation. The annual variability in the historical runoff averages 9% of the annual runoff when averaged over the entire GBNPP domain. The standard deviation can be calculated annually or monthly, by grouped or individual watershed, or over the entire domain for every variable. The reasoning for not including variability ($\sigma$) in every monthly figure is to cut down on visual clutter and also because the variability differs by a few percentage points from watershed to watershed.

Table: Historical variability (standard deviation of 36-year climatology; $\sigma$ (m)) for annual runoff, annual snow precipitation, and annual snow water equivalence (SWE) spatially aggregated for each grouped and individual watershed.

| | Annual Runoff | | | Annual Snow Precipitation | | | Annual SWE | | |
|---|---|---|---|---|---|---|---|---|---|
| **Watershed** | **(m)** | **$\sigma$ (m)** | **(%)** | **(m)** | **$\sigma$ (m)** | **(%)** | **(m)** | **$\sigma$ (m)** | **(%)** |
| GBNPP | 3.4 | +/- 0.3 | 9 | 2.0 | +/- 0.3 | 15 | 1.1 | +/- 0.2 | 18 |
| North | 3.1 | +/- 0.3 | 10 | 1.9 | +/- 0.3 | 16 | 1.1 | +/- 0.2 | 18 |
| West | 4.3 | +/- 0.4 | 9 | 2.1 | +/- 0.3 | 13 | 1.2 | +/- 0.2 | 17 |
| West-Arm | 3.4 | +/- 0.4 | 12 | 3.3 | +/- 0.5 | 15 | 1.9 | +/- 0.3 | 16 |
| East Arm | 3.6 | +/- 0.4 | 11 | 1.8 | +/- 0.3 | 17 | 1.0 | +/- 0.2 | 20 |

Additionally, Reviewer 2 asks about the uncertainty of the forecast scenario results. First, because of this comment and the comments of another reviewer, the authors chose to adopt the language of 'projection scenario', instead of 'forecast' scenario for the final draft of the manuscript. Adopting the term 'projection' is more in line with the original intent of the manuscript. As Reviewer 1 notes, "I would suggest using "projection" instead since this term acknowledges additional uncertainty involved in climate scenarios which arise from uncertain greenhouse gas emissions." We highlight our response to Reviewer 1 here: *We are not attempting to make a climate prediction (as defined in the provided weblink;* [http://glossary.ametsoc.org/wiki/Climate_prediction](http://glossary.ametsoc.org/wiki/Climate_prediction)*) of what will happen in the future in Glacier Bay. We are modeling one of the potential scenarios that may occur in the hydrology of the region that would accompany the RCP8.5 emissions scenario. For these reasons, the authors have chosen to change the term 'forecast scenario' to 'projection scenario', in every instance, throughout the manuscript. These changes are added to the final draft.* The authors have also added text in Section 3.5.1 to make this point clear in the final draft of the manuscript.

Lastly, the annual variability of the modeling results can be quantified, and the table above shows the historical variability (standard deviation) of runoff, snow precipitation, and SWE. However, there are many ways uncertainty affects the modeling results that are difficult to characterize and unrelated to annual variability. For example, spatially explicit modeling at 250m resolution is a simplification of a hydrologic system and environment that, in reality, operates at infinitely smaller scales. There are also environmental processes imperfectly described my model physics or imperfect model parameterizations that increase the uncertainty of the results. Additionally, a weather reanalysis product is used to force the model in the absence of long-term, in-situ weather station data, and there are errors and biases associated with reanalysis data assimilation and interpolation processes. The RCP 8.5 projection scenario then adds more layers of uncertainty, due to GCM model resolution and physics, and the associated likelihood of greenhouse gas concentrations in the future. Fully quantifying each of these uncertainties is not the aim of this study, and we acknowledge that this is an important limitation of the current study. We simultaneously think that since no long-term weather station records, stream flow gauges, or glacier mass balance programs exist within the study area, our physically-based, spatially explicit modeling approach is a valuable inquiry into the hydrology of Glacier Bay. It advances the current knowledge of a system that is not well characterized or measured, and it advances our understanding beyond the current literature.

We are not attempting to make a prediction of the likelihood of future hydrologic conditions, only describe the historical and projection scenario climatologies of these conditions. To address these concerns, we changed the language throughout the manuscript from 'forecast' to 'projection scenario' to clarify future scenario results, and we've added multiple clarifying sentences about the purpose and aims of the study (Abstract, sentence 3; Section 1, para 8; Section 3.5.1, para 2).

**Specific comments: 1) Title: what is hydrologic diversity? This term is never mentioned in the ms accordingly it seems misleading to use it in the title. I would be helpful to have a title that reflects the content of the ms.**

Response: Another reviewer commented on the ambiguity of this term in the title too. The title has been changed to *Seasonal Components of Freshwater Runoff in Glacier Bay, Alaska: Diverse Spatial Patterns and Temporal Change.*

**2) Abstract: An introductory sentence explaining the problematic and the purpose of the ms is missing;**

Response: The authors are comfortable leaving the abstract summarization of the research questions and problems primarily intact. Some of the language in the abstract has been changed to clarify the original intent of the manuscript. Additionally, the authors have added a single introductory sentence (in italics below) explaining the purpose of the manuscript.

*The purpose of this study is to characterize the recent historical components of freshwater runoff to Glacier Bay and quantify the potential hydrological changes that accompany the worst-case climate scenario during the final decades of the 21st century.*

**L16 why "wide variety", the same "variety" exist in any glaciated catchment;**

Response: To address this comment about a 'wide variety' vs 'variety', the authors removed 'wide' from the final version of the abstract.

**L24: this sentence is redundant, as it does not contain any conclusive information about the study:**

Response: After further review, the authors agree that this sentence is redundant, and the sentence has been clarified in the final version to reduce repeated ideas.

**3) Introduction: Nice description of the study site; however, a description of the linkage between fresh water inflow in an ocean bay and the subsequent impacts on marine life is missing; also a review of the literature of intruding freshwater into water bodies would be helpful.**

Response: The authors are attempting to balance the length of the manuscript and the scope of the ideas covered in the intro section. As previously mentioned, we added a paragraph in Section 1 to address some of the missing linkages between freshwater inflow and impacts on marine life.

**pg3, L 15: "the goals are different": it would be helpful to outline the goals;**

Response: To address this comment about the last paragraph in Section 1, and other related comments from other Reviewers, the authors added two sentences in this paragraph in the final draft to clarify the goals of the study.

**L20: the results present do not convincingly present changes in the coastal runoff (see major concerns).**

Response: The text in this line of the manuscript originally read '*The results of this study will add to the understanding developed by Etherington et al. (2007) and will provide constrained estimates of how much the coastal runoff in GBNPP will change in the future.*' After acknowledging the comments from this and other reviewers, the authors have changed this sentence to more accurately reflect the original intent of the manuscript.

The sentence now reads, '*These results will add to the understanding developed by Etherington et al. (2007) and will provide constrained estimates of potential changes in runoff in GBNPP under the RCP 8.5 projection scenario.*' This change in language is important, and the scrutiny of this

sentence by the Reviewers makes sense. We are not attempting to make a climate prediction or define the likelihood of what will occur hydrologically in the future. We are saying that if the greenhouse gas concentrations associated with the worst-case scenario RCP 8.5 come to pass, the hydrology (runoff, precipitation, SWE, etc.) will be affected in the specific ways that are outlined in our results and discussion sections. This difference is important to note and we have made an effort to clarify this throughout the paper.

**4) Methods1: pf4, L 8: model chain? Only two models are used, one for the reanalysis of the forcing data and the SnowModel.**

Response: It is true that SnowModel and MicroMet are the only two models used for this study. Therefore, the words 'model chain', and every reference thereafter, were removed from the manuscript.

**5) Methods2: I think the clarity of the ms would improve if methods and data were two separate chapter;**

Response: The methods and datasets have been presented in the same section of the publication because the input datasets are closely linked to the modeling process. The authors think the article benefits from the inclusion of these two parts in the same chapter, and it is quite common for articles in The Cryosphere to present data and methods together. However, as noted by Reviewer 3, some of our methods were first presented alongside results in various results paragraphs. The authors have removed all of the text about methods from the results and placed them into the methods sections for clarity.

**6) Methods 3: 3.4. describes in a very rudimentary way model calibration; r2 and NSE values are provided. Since the authors claim to provide an "added understanding" and "constrained estimates of how coastal runoff will change in the future" I would expect a thorough discussion on the efficiency of their modelling in regard to runoff, snow melt and ice melt contribution during the calibration period. If the calibration is not presented adequately, how can one trust in the results of future runoff?**

Response: This comment makes sense, and the original version of the calibration description was lacking in clarity and length. As previously mentioned, we added description and details about the calibration decisions to Section 3.4 in the final draft.

**7) Results: pg7,8: here results and methods to calculate the results are in the same chapter; I think a clear separation between methods and results would be helpful;**

Response: This comment is similar to other Reviewer comments about the results section occasionally mixing descriptions of the methods into the text. We have made an effort to move all of these discussions about methods to the previous section whenever necessary in the final draft.

Additionally, the authors want to clarify that the descriptions of the results through Eq. 2, Eq. 3, and Eq. 4 are not equivalent to descriptions of methods. We are attempting to clarify exactly what is represented in the results figures, so that readers can easily understand what is meant by a changes in metrics like temperature, precipitation, and snowfall equivalent to total precipitation. These metrics, and changes in their values from a historical period to a projection scenario, are not methods for producing modeling results. They are metrics that describe the results produced by the simulations. There could be many ways to describe model output in terms of temperature, precipitation, and snowfall, and we are making an effort to be very clear about how we are presenting these changes in the results section. For these reasons, the authors think the descriptions of these equations belong in the results section and not in the methods section.

**8) Figures 1, 2 and 5 (in total 8 maps) all show specific aspects of the study site; this information could be combined and presented in one or perhaps 2 large panels.**

Response: Since Figure 5 is a visualization of the results of changes in glacier coverage between the historical period and the projection scenario, the authors think Figure 5 needs to remain on its own as a depiction of glaciers in the study area.

The nested watersheds study design, depicted in Figure 2, is complex to visualize in the same map. We are presenting a study design that includes different watershed scales, the entire study area as a group (GBNPP), the grouped watersheds that flow into Glacier Bay and the Pacific Ocean, the grouped watersheds in separate arms of Glacier Bay, and the individual watersheds that lie within these other areas. Visualizing these on the same map is likely to be difficult to interpret by readers and presenting them in separate panels, side-by-side was an important decision by the authors to clearly depict the study design. Additionally, due to another Reviewer's comment, we decided to add labels for Glacier Bay, the Pacific Ocean, and Icy Strait in Figure 2.

As for Figure 1, the authors think that it is necessary to show the larger region of interest, as well as the digital elevation model/bathymetry and intend to keep Figure 1 intact.

**9) Figure 3: I do not understand why contour plots are used here; bars indicating the exact value of T and P change would be more helpful.**

Response: Color heatmaps are one of the best ways to convey hundreds of values, along multiple axes through the use of a color gradient. The authors chose to use the heatmap to visualize these changes in temperature and precipitation because it efficiently and intuitively communicates 104 different values in a compact and clear package. Since this is a 3 panel figure, the authors think

that displaying 312 bars or points representing these changes or 312 numbers in a large table would not clearly or succinctly communicate the changes in these 3 variables over time.

**10) Figures 4, 6, 7, 8, 9 and 10: it would be helpful to add an uncertainty to each point; e.g. stdev from the mean over the 30 yrs (but this would only account for climatic availability); I recommend checking recent literature on this topic.**

Response: We've chosen not to include the variability at each point for the modeling results in these figures because we want to simplify the visualization of the results. However, the authors have done some additional analysis to show the reviewers the standard deviation from the mean for some of the historical climatologies in runoff, snow precipitation, and SWE. See Table above for all of the grouped and aggregated watersheds and the authors' response to an earlier comment about uncertainty and variability.

---

## Author Comment (AC3) · 7 May 2019

Reviewer 3: Janet Curran

**Title: Can you find a term other than "hydrologic diversity" that better brings the topics of changes in runoff volume, seasonality, and drivers to mind? Hydrologic regime diversity. . .? Freshwater runoff . . .? Not sure I have the perfect term, might take a phrase to say it.**

Response: This comment is similar to another reviewer's comment about the term 'hydrologic diversity' being undefined and potentially ambiguous. The authors will change the final version to *Seasonal Components of Freshwater Runoff in Glacier Bay, Alaska: Diverse Spatial Patterns and Temporal Change*.

**Abstract, L 24-25: "a variety of changes" is vague. What is meant here?**

Response: The authors have clarified the meaning of this sentence in the final draft, which reads, "The hydrographs of individual watersheds display a diversity of changes between the historical period and project scenario simulations, depending upon…"

**P6, L15-17: The closest calibration point isn't always the most appropriate. Can you also say that the Mendenhall basin is the most similar?**

Response: See the new added paragraph in the calibration Section 3.4 that addresses this comment and the comment below simultaneously.

*Recent studies (Beamer et al., 2016; Lader et al., 2016) have investigated the accuracy and biases of the MERRA reanalysis product in coastal Alaska compared to other reanalysis products such as ERA-Interim (Dee et al., 2011), CFSR (Saha et al., 2010), NCEP-NCAR (Kalnay et al, 1996), NARR (Mesinger et al., 2006), and others. Many SnowModel parameters were tested by doing a sensitivity analysis for each reanalysis product, including monthly precipitation adjustment factors, snow/rain temperature thresholds, snow and ice albedo factors, and more (see Beamer et al. (2016) their Table 2). For each of 4 reanalysis products, they calibrated model parameters based on observations of streamflow (Q) and glacier mass balance (B). The MERRA simulation Coefficient of Determination scores (r2) for glacier mass balance (B) and stream discharge (Q) for the Beamer et al. (2016) study were 0.80 and 0.95, respectively, and the Nash Sutcliffe Efficiency (NSE) scores were 0.67 and 0.91, respectively. While Beamer et al. (2016) identified the CFSR product as the 'best overall' for the GOA region, they found that MERRA was superior at the Mendenhall Glacier observational station, which is the closest calibration point (< 25 km) to GBNPP. For these reasons, in this study we rely on the model calibration of Beamer et al. (2016; their section 3.4) and we adopt their calibration parameters for SnowModel from their Table 2 and Table A1.*

*Long-term glacial mass balance programs and long-term streamflow gauge datasets do not exist within the GBNPP study area, thus constraining our ability to conduct additional calibration*

*efforts. While the Mendenhall Glacier observation station is close in proximity to Glacier Bay, the glacier has receeded and thinned significantly since the early 1900's, glacial wastage is a significant component of annual streamflow (17%), and glacial meltwater contributes heavily to streamflow in the summer (50%; Motyka et al., 2003). As a result of these similarities in geography and hydrology, we rely on the calibration process, parameters, and best-performing reanalysis product (MERRA) from Beamer et al. (2016) for our study.*

Motyka, R.J., O'Neel, S., Connor, C.L. and Echelmeyer, K.A.: Twentieth century thinning of Mendenhall Glacier, Alaska, and its relationship to climate, lake calving, and glacier run-off. Global and Planetary Change, 35(1-2), pp.93-112, DOI: blah, 2003.

**P6, L17-18: These metrics are for the Beamer et al. (2016) study, correct? Since this "calibration" section oddly refers to the calibration of a prior study, I suggest phrasing this clearly so the skimming reader doesn't assume these metrics are for your study.**

Response: These metrics are from the Beamer et al. (2016) study. This paragraph in Section 3.4 has been re-written to clear up any vague phrasing that may exist for the final draft. The authors have also added some additional explanation of the calibration process for clarity due to the comments of at least one other Reviewer on the calibration process. See new, additional calibration paragraph above.

**P8, L1: This sounds like a justification for a higher-than-expected result, but the wording isn't clear. Does the RCP8.5 scenario establish a minimum of 3 degrees change?**

Response: The RCP8.5 scenario, and corresponding SNAP temperature and precipitation anomalies, do not establish a minimum or baseline of 3 degrees change. The statement in the text is simply a description of the lower range of temperature changes that are found within each watershed group between the historical and projection scenario modeling results. In Section 4.1 we changed the end of this sentence to avoid confusion.

**P10, L3: Is the 3.40 m/yr value actually for "runoff", not "precipitation" as stated? That would be more consistent with the value in the next line.**

Response: In this case, for all watersheds in the GBNPP domain, the annual average historical precip value is 3.4m and the annual average historical runoff value is 3.4m. This is different within each grouped watershed. Remember, the authors are reporting the runoff value calculated by SnowModel, which includes all the water made available at each grid cell from precip (snow & rain) + glacier melt processes. ET is not subtracted from these runoff values, just simply plotted on the same graph for context (from Figure 7 and Appendix B). We also do not estimate long-term changes in the groundwater or glacial wastage (ΔStorage) within the 36-year historic and 30-year projection scenarios. For a further discussion of ET calculations and estimations, please refer to the responses to Reviewer 1.

Let's look more in depth at some of these runoff and precipitation values. See the table below for additional information. When PRECIP < RUNOFF like in the West watershed below, there is likely glacier ice melt occurring over the time period that is supplementing the freshwater flows in the basin. When the snowpack disappears from a glacier grid cell, the energy balance model melts the glacier ice and adds to water to the runoff variable. When PRECIP > RUNOFF, like the West-Arm and Tarr watersheds below, there is likely SWE left over at the end of the the year, on glacier surfaces and at the highest elevations, that gets zeroed out at the end of the water year in the SnowModel simulations. This is because SnowModel does not include glacier dynamics, and there will be net accumulation of snowpack above the equilibrium line. When running multi-year simulations, SnowModel offers the option to zero out all the grid cells that still contain SWE on the last day of the water year, in order to not carry over SWE into the next water year. In this study, we chose to zero out the SWE in this manner at the end of every water year, and this method is the recommended/default method for SnowModel users when running multi-year simulations in regions with glacier coverage. In watersheds like the West-Arm and Tarr, where there is 54%-68% glacier coverage, the amount of snow precipitation and leftover SWE is likely to be a substantial portion of the overall precipitation, and that is reflected in the Runoff and Precip values for these watersheds. However, we have not conducted the spatial or temporal end-of-water-year SWE analysis for all years and all watersheds because it is not directly related to the aims and scope of this study.

When PRECIP ~ RUNOFF, like in the GBNPP aggregated watershed, there is a balance between these glacial melt processes and leftover SWE at the end of the time period.

| Watershed | Historical PRECIP (m) | Historical RUNOFF (m) |
|---|---|---|
| GBNPP | 3.4 | 3.4 |
| North | 3.3 | 3.1 |
| West | 3.8 | 4.3 |
| West-Arm | 4.4 | 3.4 |
| East-Arm | 3.3 | 3.6 |
| Tarr | 6.1 | 2.9 |
| Carroll | 2.4 | 2.9 |
| Dundas | 2.4 | 3.1 |

**P12, 13-14: This statement about a non-stationary system is inconsistent with the presentation of Figure 10 on P10, L20-22, which notes little significant change with one basin**

**as an exception. The trend for the excepted basin isn't very convincing (p > 0.05, short and discontinuous dataset, a bit noisy), making the comments on P12 seem overstated.**

Response: Based on the current Figure 10, your observation and comment is warranted. This is an overstatement in this context. The sentence was originally written for a different figure showing more significant trends in the grouped GBNPP watershed overall, but we had previously decided to remove that figure and the previous results. We will adjust the wording accordingly, by taking out the reference to non-stationarity, and the authors have also decided to remove Figure 10, because the presence of this figure is not adding any additional, necessary information to the manuscript. We've adjusted the language in Section 4.6, paragraph 2.

**Appendix B: Interestingly, the forecast runoff hydrographs, which admirably show the relative contributions of runoff processes, produce a few seasonalities that aren't apparent for individual streams in my present work characterizing historical hydrographs. The composite GBNPP and North basins appear to have a snowmelt-dominated spring peak and a larger rainfall-dominated fall peak, a reversal of the typical relative magnitudes for a bimodal glacierized basin hydrograph. Can this be explained by an increase in spatial distribution of future rainfall-dominated areas within the composite basin or any other observations from the modeling?**

Response: This is an excellent question, and one that would require more in depth spatial analysis of the modeling results to answer quantitatively. At first glance, your suspicion about the increase in spatial distribution of future rainfall dominated areas within the composite basin contributing to the larger rainfall dominated fall peak makes sense in the context of the temperature and precipitation results. Looking at Figure 3 may shed some light on this question. We can see the largest increases in temperatures, the largest increase in precipitation, and the most significant decreases in snowfall occur for most watersheds during the months between Oct and Dec (appearing as a blob of darker colors during those months). However, to calculate the exact spatial extent (area in $km^2$) of the increase in rainfall-dominated areas would require additional spatial analysis of the results during both the historic and projection scenarios. Since a close look at Figure 3 contains some of the answers to your questions, the authors are comfortable keeping the Appendix A explanation in the manuscript and figure caption without changes.

Additionally, the freezing line altitude (FLA) analysis (see Figure 11) includes a ΔFLA from historic to projection scenario for the winter (+234 m) and summer (+1341 m) months. We did not include the spring (+910 m) and fall (+775 m) ΔFLA values in the figure due to simplicity's sake. These increases in the FLA during spring and fall would represent a large corresponding increase in the spatial distribution of the rainfall dominated areas in the RCP8.5 projection scenario, and makes sense with the changes in temperatures found in Figure 3.

**Introduction: Trim and keep focused on the study by omitting details about GOA (especially in 1st and 2nd paragraph), minimizing drama (P2, L6-7), and considering moving setting**

**information to the Study Area section if it's actually needed (the long discussion of tidal mixing and stratification and the Etherington et al. study made me think this was the study focus on first read). It's all interesting, but it's not until the penultimate paragraph (P3, L37-39) that the problem is hinted at and not until the final paragraph (P3, L13) that the actual work of the study is introduced, and the reader can finally start understanding the direction of the manuscript.**

Response: The discussion of the GOA in the intro paragraphs 1 & 2 are important to give the readers of the Cryosphere the geographical context within which the Glacier Bay region exists. However, some of the details and general statements about the region can be trimmed to keep the focus on Glacier Bay.

In Section 1, para 2, the word 'dramatic' will be changed to 'considerable'. Also the authors are comfortable taking out the end of the following sentence, which will now read 'Indeed, the coastal mountain ranges of Alaska have recently sustained rapid rates of deglaciation (Arendt et al., 2002; Arendt et al., 2009; Gardner et al., 2013; O'Neel et al., 2015).' Drama was not the intent with the previous sentence and wording, and these changes remove any question or potential of dramatic language.

Reviewers have requested both more information (Reviewer 1) about oceanographic processes (stratification, mixing, etc.) and less information about these processes (Reviewer 3). The authors are attempting to balance all the requests, while simultaneously keeping the original flow and intent of the article. In this case, we choose to keep the remaining paragraph structure of the intro intact.

**P2, L31 and P3, L3-4: The number of references to particular places within Glacier Bay suggests Figure 2 could be presented earlier.**

Response: This is a good suggestion. Figure 2 will now be referenced in this paragraph.

**Study Area, paragraphs 1 and 2: Clearly define study area (all watersheds within GBNPP, which includes all the lands of GBNPP and some areas outside it?). The multiple nested, paired watersheds are a nice study design but are hard to keep track of. Suggest moving the parts of paragraph 2 that aren't obvious from the figure or table (P3, L29-30) into paragraph 1. Consider using a defining characteristics for the names or adding a column to Table 1 to associate basin names with a defining characteristic. It would be helpful to know "North" is the full Glacier Bay basin and that the choice of the three named basins allows comparison of basins having. . .(a range of elevation? a range of glacier characteristics?), for example.**

Response: The authors have added text in this paragraph to clarify the study area boundary decisions in Section 2. In this section, para 1 describes the entire GBNPP study area, and para 2 describes the 4 grouped watersheds, and paragraph 3 describes the individual watersheds. The

authors think this is an appropriate structure for the paragraphs, and some additional text has been added to these paragraphs to clarify.

**P7, L23-24: This is one of the clearest statements of the goals/outcome of the study. Could use this earlier.**

Response: This is a good suggestion from Reviewer 3. This sentence has been altered slightly and added to the last paragraph of the Section 1, in order to more clearly state the outcome of the present study.

**P9, L11-12: Delete information repeated from methods.**

Response: This is another good suggestion that makes the manuscript more concise. The first sentence in Section 4.3 summarizes paragraphs found in the methods section and has been removed from the final draft.

**P9, L25-35: Many details of computations, and the discussion of the omission of routing, seem like methods. Consider moving to Section 3.1 or elsewhere in Methods.**

Response: Again, this comment from Reviewer 3 makes sense for the flow of the paper and for keeping a clear delineation between methods and results. This paragraph has been moved to the methods section and removed from this results paragraph in the final draft.

**P10 and 11, Section 5, first and second paragraphs: Most of the main points are made in the first paragraph; suggest combining the two and reducing detail. Consider moving computation of FLAs to methods.**

Response: The authors think the detail in this paragraph is a warranted discussion of the FLA analysis and the landscape dependencies of seasonal snow patterns. However, there are a few sentences in this section that have been appropriately moved to the methods section in accordance with this suggestion.

**P12, L5-6: Nice explanation of why CTD dataset was included, could use this earlier.**

Response: This is an important point, and the authors have restated this in Section 1, para 7.

**P11, L11-12 and L20: Check figure number. I assume you mean figure 11a and b, respectively.**

Responses: Good catch, yes, this was previously Figure 9 and was changed in the final stages of drafting.

**References: References are used appropriately. I did not check to make sure all are used, or that all references cited are included. The recommended citation for USGS reports includes the report series title and report number. For Curran et al. (2003) that's Water-Resources Investigations Report (or WRIR, if preferred) 03-4188 and for Wiley and Curran (2003) it's Water-Resources Investigations Report 03-4114.**

Response: These two citations have been changed to include the Water Resources Investigations Report.

**Fig. 1: Labeling Alaska and Canada (a) and Glacier Bay (b and c) would help reader comprehension.**

Response: Agreed, the labeling of Glacier Bay is a missing piece of these maps and they will be added to the final figures. As for labeling Alaska and Canada, the authors want to keep the maps as simple as possible, without clutter, and do not think it is necessary to know exactly where the international boundary lies, since we have no other non-physical geographical locations (cities/towns, boundaries, roads, etc.) labeled.

**Fig. 2: Label Glacier Bay. The Alaska/Canada boundary is referenced in the text but not shown here.**

Response: Again, the authors think it's worth mentioning the boundary in the text but it is not a necessary part of the maps. Glacier Bay has been labeled in the final draft figure.

**Fig. 5 : Shading of forecast glaciers is distractingly similar to ocean. The title "Glacier Change" doesn't match the legend items, which include two glacier positions and the GBNPP boundary.**

Response: For the final draft, the legend title 'Glacier Change' has been removed completely from the legend of the figure. The authors agree that the color palette for the forecast glaciers needs to be different for reader comprehension, and the final draft of the figure has a different color palette.

**Fig. 6: Suggest being consistent with the x-axis scale used for other monthly plots (use Jan-Dec, not Sept-Aug)**

Response: The reason the SWE climatologies are presented in water year format, from Sept-Aug, instead of presenting them in the previously used calendar year format is because the progression of snow water equivalence is simpler, and possibly more intuitive to visualize and understand when the winter is not split in two parts. Often, but not always, when SWE climatologies are presented in the literature they span the water year and not the calendar year. For these reasons, the authors think the SWE climatology is easiest to understand in its current format.

**Fig. 7 caption, last sentence for (a): Check for typo in "the modeled for runoff climatology"**

Response: This typo has was changed in the final draft.

**Tables 1 and 3, and Appendix A and B: Suggest some structure to convey basin/sub-basin relationship and the various pairings of nested basins (a line or spacing, for example). At a minimum, keep the same order in the Appendices as is used for the tables.**

Response: The authors have divided the tables into groupings, and the order of the Tables and Appendices are kept the same in all instances.

**Appendices: These plots are useful results and would lend themselves well to being reduced in size. Consider rearranging to fit each Appendix on 1 page with a single legend for each and including in the text.**

Response: The authors are open to including the appendices in the main text, and to reducing them in size, instead of including these sets of figures as appendices. This decision can be ultimately be left up to the editors discretion at the Cryosphere.